



# Quantifying and attributing the role of anthropogenic climate change in industrial-era retreat of Pine Island Glacier

Alexander T. Bradley[1], David T. Bett[2], C. Rosie Williams[2], Robert J. Arthern[2], Paul R. Holland[2], James Byrne[2], and Tamsin L. Edwards[1]

[1]Department of Geography, King's College London, London, United Kingdom
[2]British Antarctic Survey, Cambridge, United Kingdom

**Correspondence:** Alexander T. Bradley (alex.bradley@kcl.ac.uk)

**Abstract.** The West Antarctic Ice Sheet (WAIS) has undergone rapid change over the satellite era, characterized by significant thinning, grounding line retreat, and mass loss. Over one-third of the ice loss from this region is from Pine Island Glacier (PIG). However, robust causal links between anthropogenic climate change and PIG ice loss have yet to be established. Here we quantify the role of anthropogenic climate change in observed retreat of PIG over the 20th century and how this may evolve up

to 2200. To do so, we use an ensemble Kalman inversion data assimilation method embedded into the calibrate-emulate-sample (CES) uncertainty quantification framework. This procedure yields observationally constrained probability distributions of both model and climate forcing parameters. Our analysis suggest that it is unlikely that the extent of 20th century PIG retreat would have taken place without anthropogenically driven trends in ice-sheet forcing and that anthropogenic forcing exaccerbated the extent of PIG retreat over the 20[th] century, by approximately 18%. We also find significant retreat even with no anthropogenic

trends in forcing, potentially highlighting the role of ice-sheet memory associated with long, slow retreat over the Holocene in controlling present retreat of the WAIS. We further find that significant anthropogenic signals in climate forcing only emerge in the middle of the 22nd century. An important caveat to this work is our choice of initial state, which is larger than expected in practice and may render the anthropogenic forcing contribution in our simulations to be smaller than it is in practice.

## 1 Introduction

The West Antarctic Ice Sheet (WAIS) has undergone rapid change over the satellite era, characterized by significant ice acceleration (Mouginot et al., 2014), thinning (Smith et al., 2020), grounding line retreat (Rignot et al., 2014), and mass loss (Rignot et al., 2019). Almost all of the ice loss from the Antarctic Ice Sheet over the satellite era has come from the WAIS (Otosaka et al., 2023). Of this, a majority comes from two major outlet glaciers: the Pine Island and Thwaites Glaciers (Morlighem et al., 2020).

Despite these changes occuring at the same time as anthropogenic climate change, robust causal links between climate change and West Antarctic Ice Sheet loss have yet to be established (Meredith et al., 2019; Fox-Kemper et al., 2021). Several lines of evidence point to contributions to retreat from different sources; these include long, slow retreat since the Last Glacial





Maximum, a period of anomolously high forcing in the 1940s which triggered rapid ice loss over the 20th century, internal ice and ocean feedbacks, and anthropogenically driven changes in climatic forcing. We first briefly document this evidence.

## 1.1 Evidence linking WAIS retreat to anthopogenic forcing

Geological and geophysical evidence implies that at the Last Glacial Maximum (around 25,000 years ago), the West Antarctic Ice Sheet extended close to the continental shelf edge along the Amundsen Sea and Bellingshausen Sea margins (Larter et al., 2014). By around 10,000 years ago, grounding lines in the region had retreated significantly, approaching locations close to the present day (Larter et al., 2014). This retreat, on the order of 500 km over 15,000 years (i.e. approx 33 m per year on average), is much slower than present day retreat rates, which are on the order of several hundred metres per year (Chartrand et al., 2024). Since then, the ice-sheet geometry has remained broadly similar, albeit with localised retreat (Larter et al., 2014), which has accelerated in recent decades. Owing to their large size and high viscosity, ice-sheet response timescales (the timescale on which an ice-sheet responds to changes in climate forcing), are very long, potentially on the order of tens of thousands of years. As such, memory of the slow retreat of WAIS since the last glacial maximum may plausibly still be playing a role in its evolution today.

Sediment records from beneath the Pine Island (Smith et al., 2017) and Thwaites (Clark et al., 2024) Glaciers indicate that the present phase of rapid retreat was initiated in the 1940s. For Pine Island specifically, before this period, the ice sheet was stably grounded on a prominent seabed ridge, with the ice shelf finally detaching from the ridge in the 1970s (Jenkins et al., 2010; Smith et al., 2017). Once initiated, a range of ice and ocean feedbacks would have helped to sustain its retreat; these include: grounding-line retreat towards a deeper bed (Favier et al., 2014), increasing ice damage (Lhermitte et al., 2020), increased access of warm water into sub-ice-shelf cavities (De Rydt et al., 2014), retreat of the ice front (Bradley et al., 2022), increased ice base area exposed to warm ocean water (Holland et al., 2023), and spinup of circulation inshore of the ridge (De Rydt and Naughten, 2024). There is evidence that, following its complete unpinning, Pine Island underwent a period of irreversible retreat until the 1990s (Reed et al., 2024a, b). However, the present phase of retreat cannot be purely self-sustaining because ice discharge remains responsive to ocean forcing (Christianson et al., 2016; Jenkins et al., 2018).

During the initial retreat in the 1940s, high pressure and wind over the Amundsen Sea Embayment (Schneider and Steig, 2008) are thought to have driven increased warm Circumpolar Deep Water (CDW) access to ice shelf cavities and therefore increased ice shelf basal melting (Steig et al., 2012). Increases in basal melting reduce the restraining back-pressure ('buttressing') that ice shelves apply to the adjacent ice-sheet (Shepherd et al., 2004; Pritchard et al., 2012). O'Connor et al. (2023) demonstrated that, although the pressure and wind anomalies in the 1940s are exceptional in the context of the past century, they are not exceptional in the context of the last 10,000 years: an event of this magnitude would have occurred tens to hundreds of times over the last 10,000 years. Thus, the 1940s event is unlikely to be solely responsible for the current phase of retreat: events of this magnitude have taken place many times before, but have not initiated ongoing rapid centennial scale retreat.

This raises the question: why did the increase in CDW in the 1940 event initiate retreat? And, to what extent were anthropogenic trends in forcing responsible? Proxy-constrained climate simulations suggest that westerly winds at the Amundsen Shelf break were changing over the 20th century (Holland et al., 2019), but the anthropogenic (rather than internal) component



of this trend only emerged in the 1960s (Holland et al., 2022). Trends in westerly winds over the Amundsen Sea are a well established response of the Southern Hemisphere climate to anthropogenic emissions (e.g. Arblaster and Meehl, 2006). In ocean simulations, these wind changes drive CDW under the ice shelf cavities more frequently on decadal timescales, increasing
ice shelf melting (Naughten et al., 2022). One plausible narrative is that WAIS retreat may have been triggered in the 1940s by naturally occuring climate change, but was then sustained by anthropogenic climate forcing since the 1960s. Without this change, WAIS might have recovered, as it appears to have done many times before in the past (Holland et al., 2022; O'Connor et al., 2023).

## 1.2   Challenges in attributing West Antarctic Ice Sheet retreat

In this study, we aim to quantify the role of anthropogenic trends in forcing on retreat of the Pine Island Glacier (PIG). We focus on PIG specifically (rather than the whole of the WAIS) because of the availability of grounding line constraints prior to the satellite era (Smith et al., 2017) and because using a smaller domain reduces computational expense in the large ensemble of model simulations required. As in other attribution studies (e.g. Otto, 2017), we compare simulations of PIG response under all climate forcings with simulations under only natural forcings, aiming to reproduce the behaviour of the glacier since 1750.
We take a probabilistic approach, as in other studies, to account for uncertainties in climatic forcing, model parameters, and future forcing scenarios (Bradley et al., 2024b).

There are three challenges to address. First, simulations of ice sheet change are sensitive to choices of model parameters, some of which are very poorly-constrained. This makes reconstructing the magnitude and timing of PIG retreat challenging because only a small subspace of model parameter values are compatible with the observed retreat (discussed further in 'Meth-
ods'). Simulations in idealized geometries indicate that ice sheet retreat attribution assessments are highly sensitive to the choice of fixed model parameters (Christian et al., 2022). Parametric uncertainty (that arising from poorly constrained model parameters) must be explicitly included in attribution assessments (Bradley et al., 2024b).

Second, evaluation of past ice sheet simulations is usually limited by a lack of data before the satellite era (i.e. around 1979). Here, our focus on PIG allows us to use the sediment-based records of GL location change from the 1940s to 2010 (Smith
et al., 2017) to constrain the ice sheet model parameter distributions.

Finally, changes in Southern Ocean since 1750 are very poorly-known. We therefore use an ensemble of ocean forcing realisations, as in previous studies (Naughten et al., 2022, 2023), to include this aleatoric uncertainty (Robel et al., 2019; Aschwanden et al., 2021). The anthropogenic trend in ocean forcing is treated as an unknown parameter to be inferred in the attribution study: in this way, we learn what the anthropogenic forcing must have been in order to reproduce the past.
The issue of sensitivity of ice sheet model behaviour to poorly constrained model parameters has led to the use of observations of past ice sheet change to constrain model parameters, typically using Bayesian weighting (e.g. Ritz et al., 2015; Nias et al., 2019, 2023; Bevan et al., 2023; Wernecke et al., 2020; Coulon et al., 2024) or history matching (McNeall et al., 2013; DeConto and Pollard, 2016; Edwards et al., 2019). Essentially, the idea is to sample model parameters and then simulate ice sheet behaviour over the historical period using each of these. The parameter sets are then calibrated, either by weighting them
(in the case of Bayesian weighting) or by ruling out certain combinations (history matching) based on their agreement with





observations over the historical period. When combined with model-emulation (constructing a computationally cheap surrogate emulator of the expensive simulator), observationally-constrained distributions of model parameters can be obtained (e.g. Aschwanden and Brinkerhoff, 2022; Rosier et al., 2024; Jantre et al., 2024; Berdahl et al., 2020). One issue with such parameter sampling is that when the likely space (that is the region of parameter space where model output agrees with observations) is small, it may be only sampled with a small number of simulations, or not at all. This is particularly pertinent when large numbers of model parameters are involved or systems are non-linear (ice-sheet models have both of these features).

In this paper, we overcome this sampling challenge by using an ensemble Kalman Inversion (EKI) sampling strategy, within an uncertainty quantification framework–'calibrate-emulate-sample' (outlined in 'Methods'). Rather than selecting all parameter samples at the outset (i.e. before any simulations are performed, as is typical in ice-sheet modelling, for example using Latin hypercube sampling), the EKI is an iterative method: model parameters sampled are updated iteratively based on agreement between model simulations and observations. The EKI also benefits from being a gradient-free optimization method: it is not necessary to compute the gradient of the model–observable error with respect to model parameters (Garbuno-Inigo et al., 2020; Kovachki and Stuart, 2019) in order to perform the update. The EKI has been successfully applied to calibrate model parameters in climate models (e.g. Dunbar et al., 2021; Mansfield and Sheshadri, 2022; King et al., 2024), but we believe this is its first application in ice sheet modelling. Ensemble Kalman *filters* (Gillet-Chaulet, 2020; Choi et al., 2025) have been applied to estimate the *state* of an ice sheet model; this is in contrast to the EKI, whose purpose is to estimate model parameters. Following the EKI (the calibrate step of CES), we emulate model outputs and perform Markov chain Monte Carlo sampling to obtain posterior distributions of parameters (Cleary et al., 2021) and therefore probabilistic ice sheet reconstructions.

## 1.3 Aims and outline

There are two main aims of this paper. Firstly, to obtain observationally-constrained distributions of parameters using CES, and thus reconstruct the statistics of PIG retreat over the industrial period. Secondly, to quantify the role of anthropogenic trends in forcing in the observed retreat of the Pine Island Glacier over the 20th century, and assess how these trends might evolve in the future.

This paper is structured as follows: in section 2, we outline the methods in two parts: in the first (section 2.1), we outline the model setup, discussing the climate and model parameters which are calibrated, and the initial state from which the simulation begins. In the second (section 2.2), we describe how posterior distributions of model parameters are obtained, outlining each of the steps of the CES procedure. In section 3, we present the observationally-constrained posterior distributions of climate and model parameters and present our attribution assessments, obtained using an ensemble of simulations based on these posterior distributions. These simulations cover a time period from 1750 to 2200, allowing us to reconstruct the evolution of the role of anthropogenic forcing in the past, as well as describing how it might evolve in the future. In section 4, we describe the implications of, and uncertainties associated with, our results. In section 5, we summarise our results.



## 2   Methods

### 2.1   Model and Climate Forcing Parameters

We simulate the retreat of the PIG over the industrial era using the WAVI ice-sheet model. A full model description can be
found in Arthern et al. (2015); Bradley et al. (2024a); here we note several features of this model, which are relevant to our
model parameter choices. Specifically, we consider three poorly constrained model parameters: a basal sliding prefactor, an ice
viscosity prefactor, and an ice-shelf basal melt rate exponent prefactor. Observationally-constrained posterior distributions of
these three parameters, alongside a further three climate forcing parameters (see section 2.1.3), are determined using the CES
procedure (see section 2.2). In this section, we describe how these model and climate parameters enter into our model and its
initial state.

#### 2.1.1   Ice sheet model and parameters

In WAVI, the ice viscosity $\eta$ is related to the ice flow velocity components $(u, v, w)$ by (Goldberg, 2011)

$$\eta = \frac{(A_0 A)^{-1/n}}{2} \left[ u_x + v_y + u_x v_y + \frac{1}{4}(u_x + v_y)^2 + \frac{1}{4}u_z^2 + \frac{1}{4}v_z^2 \right]^{\frac{1-n}{2n}}, \tag{1}$$

where subscripts denote partial derivatives with respect to Cartesian co-ordinates $(x, y, z)$, and $n = 3$ is the Glen coefficient.
Here, $A$ is a 'Glen A coefficient' which premultiplies the . $A_0$ is a spatially constant, order one prefactor which allows the ice
viscosity to be tuned (note that higher $A_0$ corresponds to lower viscosity). The field $A$ is determined using an inverse method,
as outlined in Arthern et al. (2015); this procedure yields the field $A$ and a bed friction field (see below) which minimise the
misfit between modelled and observed ice velocity and surface elevation rate-of-change fields. We use MEaSUREs 2014/2015
for surface velocities (Mouginot et al., 2017), surface elevation change from Smith et al. (2020).

WAVI applies a Weertman-type sliding law, in which the basal shear stress, $\tau_b$, is related to the basal sliding speed, $u_b$, by

$$\tau_b = C_0 C |u_b|^{1/m}. \tag{2}$$

Here, $m = 3$ is a Weertman sliding exponent, and $C$ is the spatially variable field of basal sliding coefficient from the inversion.
$C_0$ is an order 1 prefactor, which permits the precise basal sliding field to be tuned, and is referred to here as a 'basal sliding
prefactor'.

As outlined in the introduction, since the 1940s, PIG has retreated from a prominent seabed ridge (Figure 1a-b). Therefore,
there are areas of our model domain in which the ice is grounded in the past, which are not grounded in the present day
geometry. In these areas, the inversion, which is based on present day geometry, returns a basal sliding coefficient of zero
(there is negligible friction at the ice shelf-ocean interface). It is not possible to invert for bed friction at the time when the ice
was grounded in these areas (i.e. prior to the 1970s) because the satellite records do not extend this far back. Following Reed
et al. (2024a) we set the basal sliding coefficient $C$ in these regions to be a constant value of 10,000 Pa m$^{-1/3}$ a$^{1/3}$. Note that the
basal sliding prefactor also pre-multiplies the drag in these areas and therefore the basal shear stress there is also tunable.

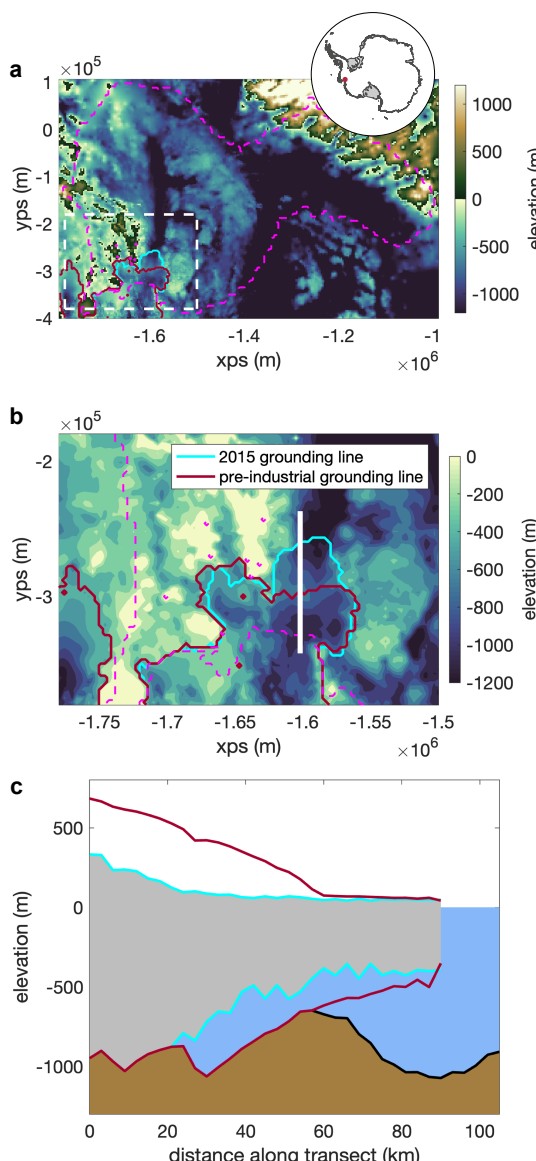

**Figure 1.** (a) Contour plot of the bed beneath Pine Island Glacier (colours) alongside the outline of the ice mask (pink dashed line). The cyan and red lines indicate the grounding line position in 2015 and in the steady state pre-industrial configuration, respectively. The inset shows the location of Pine Island Glacier in Antarctica as a red point. xps and yps are co-ordinates in the polar stereographic projection. (b) As in (a), but zoomed in on the Pine Island Ice Shelf (the white dashed box in (a)). The thick white line indicates the centreline along which the transects in (c) are taken, and along which grounding line retreat is measured. (c) Along centreline bed topography (shaded brown) alongisde ice geometry in 2015 (shaded grey, cyan outline) and in the pre-industrial steady state (red).





The WAVI model comes equipped with a suite of melt rate parametrizations (Bradley et al., 2024a). Here, we apply a quadratic parametrisation of basal melting, where the basal melt rate on floating ice shelves is

$$\dot{m} = MT_*^2. \tag{3}$$

Here $M = 5.0 \times 2^{M_0}$ m $°C^{-2}$ $a^{-1}$ is a variable melting calibration coefficient, with $M_0$ referred to as the melt rate exponent prefactor, and

$$T_* = \lambda_1 S_a + \lambda_2 + \lambda_3 z_b - T_a \tag{4}$$

is the local thermal forcing on the ice shelf. $T_a$ and $S_a$ are, respectively, the local ambient temperature and salinity (see section 2.1.3) and $z_b$ is the basal elevation, all evaluated at the local ice-shelf base in each location. Here, $\lambda_1 = 5.73 \times 10^{-2}$ $°C$ is the

liquidus slope with salinity, $\lambda_2 = 8.32 \times 10^{-4}$ is the freezing temperature offset, and $\lambda_3 = 7.61 \times 10^{-4}$ $°C$ $m^{-1}$ is the liquidus slope with depth.

     We apply a temporally constant surface accumulation field (Arthern et al., 2006) across our domain, set to present day values.

     We simulate the retreat of the PIG on a numerical grid with 3 km resolution. This resolution is chosen as a balance between retaining accuracy and permitting large ensembles of simulations to be run. Recent work (Williams et al., 2025) has shown that

numerical simulations are not overly sensitive to resolution in WAVI, provided they are at or below 3 km.

     In all simulations, we use a fixed time-step of 0.05 years. However, the Courant-Friedrichs-Levy (CFL) condition, which determines numerical stability, is model parameter dependent. When model parameters are varied in the EKI procedure, we therefore occasionally encounter numerical instabilities. These instabilities manifest as 'stripy' ice velocities and thicknesses, particularly in the fast flowing regions of the Pine Island Ice Shelf. Comparison with simulations in which the timestep is

reduced reveal that, despite this, modelled ice volume change and grounding line retreat (which we use as observational constraints on our model – see section 2.2.2) are not affected by this. In addition, simulations which violate the CFL condition are typically associated with rapid retreat (e.g. they correspond to very low basal friction), much faster than observations suggest. Therefore, these simulations display poor agreement with observations and do not contribute significantly to model parameter estimation.

**2.1.2   Initial state**

To reproduce the ice sheet geometry prior to the satellite era, we follow Reed et al. (2024a): we first take the present day PIG state (figure 1a-b) and advance the grounding line to the seabed ridge on which it was grounded prior to the 1940s (Smith et al., 2017). This is achieved by turning melting off entirely and timestepping the model forwards for 500 years. After this time, the ice sheet is stably grounded on the ridge with an approximately constant grounding line position and ice volume. In

this configuration, the grounding line position is consistent with sediment records (Smith et al., 2017), but the ice volume is too large (figure 1c; we do not have observational constraints on the ice geometry prior to the satellite era, but at least *some* ice shelf basal melting must have been taking place). To address this, we run a spin-up period in which melting is turned on, with $M_0 = 0$, and cold ocean forcing conditions (corresponding to the coldest observed conditions, defined formally in section



2.1.3) applied. With $M_0 = 0$, basal melt rates in the present day geometry yield total freshwater fluxes that closely match
observations (Dutrieux et al., 2014). This spin-up period lasts 50 years; during this time, the grounding line retreats slightly,
but remains grounded on the ridge.

The ice geometry after the spin-up is considered to be the pre-industrial state. Each of our simulations begins in 1750. Ideally,
we would run our simulations for longer, by setting this state to correspond to further back in time, allowing memory of the
initial state to be lost. However, this incurs an additional computational expense, and 1750 is chosen as a balance between extra
computational expense and running for long enough to minimise initial state dependence. We discuss the implication of this
choice in section 4.

### 2.1.3 Climate Forcing and Parameters

Following the spin-up, variable climate forcing is applied to the ice sheet via the ambient temperature and salinity, $T_a$ and $S_a$, in
the melt rate parametrization (3). We take ambient temperature and salinity profiles which are similar to observations (De Rydt
et al., 2014; Dutrieux et al., 2014). These profiles are piecewise linear: they are constant in both an upper (temperature -1°C,
salinity 34 PSU, corresponding to Winter Water) and lower layer (temperature 1.2°C, salinity 34.7 PSU, corresponding to
Circumpolar Deep Water), which are separated by a pycnocline of 400 m thickness, across which the temperature and salinity
vary linearly (figure 2a-b). Ocean variability and trends in the Amundsen Sea are primarily manifested as a thickening or
thinning of the deep Circumpolar Deep Water layer, rather than substantial variation in the temperature or salinity of the
upper or lower layers (Dutrieux et al., 2014; Naughten et al., 2023). Therefore we impose all climatic variability by raising
and lowering the pycnocline, and so the depth of the centre of the pycnocline, denoted $P_c$, parameterizes the entire ambient
temperature and salinity profile.

The temporally varying pycnocline centre depth encodes information about climate forcing. It is expressed as term corre-
sponding to internal variability super-imposed on terms corresponding to the 1940s event and anthropogenic trends in forcing,
with the latter two representing the theorised leading order controls on retreat over the industrial era. These are encoded by
setting the pycnocline centre as (figure 2c):

$$P_c(t) = P_{c,0} + \alpha \frac{R(t)}{4} + B(t) + T(t) \tag{5}$$

Here, $P_{c,0} = -500$m is a typical observed pycnocline depth (Webber et al., 2017). The second term on the right hand side
of (5) corresponds to internal variability. The shallowest and deepest observed pycnocline depths are approximately -400m
and -600m, respectively (Dutrieux et al., 2014) and thus the parameter $\alpha = 200$ m is the magnitude of differences between
warmest and coldest conditions associated with internal variability. $R(t)$ is a dimensionless timeseries generated from a modi-
fied first-order autoregressive process: it is as in Christian et al. (2022) and Bradley et al. (2024b), with interdecadal-to-decadal
timescales well represented, but capped between -2 and 2. Note that this formulation assumes that the pre-industrial pycnocline
centre is equivalent to present day; this is consistent with an assumption of zero anthropogenic trend in forcing, which we take
as the prior value on this quantity (see section 2.2.1). In the spin-up period described in the previous section, the pycnocline
centre is set to a constant value of -600m.





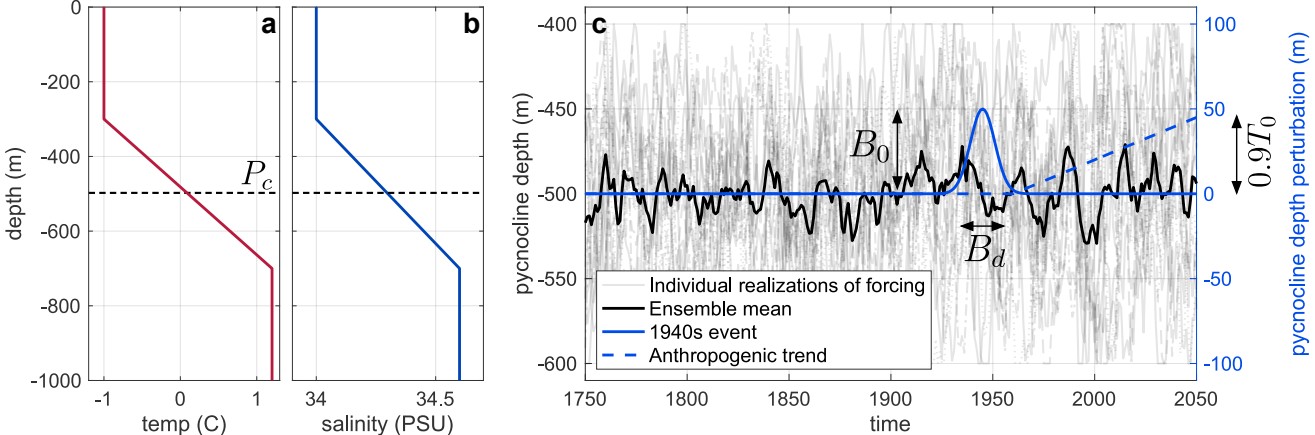

**Figure 2.** Example ambient temperature (a) and salinity profiles (b) used in the melt rate parameterization (equations (3)–(4)). The pycnocline is displaced up and down to represent climatic variability, with no change in upper or lower layer properties, so the depth of the pycnocline centre, $P_c$, parametrizes the profiles. In the examples shown here, $P_c = -500$ m. (c) Components of the forcing (equation (5)). Translucent black curves indicate individual realizations of internal variability, $P_{c,0} + \alpha R(t)$, and the solid black curve indicates the ensemble mean of these. Also shown (corresponding to the right-hand axis) are the 1940s event perturbation to the forcing, $B(t)$ (solid blue), and anthropogenic trend, $T(t)$ (dashed blue). In this case, we show $B_d = 5$ years, $B_0 = 50$ m, $T_0 = 50$ m ($0.9T_0$ is indicated because the centennial trend is initiated in 1960 and plotted until 2050, i.e. 90 years).

To facilitate a quantification of aleatoric uncertainty (uncertainty associated with the chaotic forcing), we repeat the CES procedure outlined in the following section for multiple realizations of the autoregressive process, which we refer to as realizations of forcing. In total, we consider 14 individual realizations of forcing, shown in figure 2d. The ensemble mean of the autoregressive internal variability components is close to, but not zero (figure 2d), owing to the finite size of the ensemble of different realizations.

Since we know that the 1940s event is an important piece of the puzzle, we explicitly superimpose a term capturing this behaviour onto the internal variability, which does not capture the anomalously strong 1940s event in the ensemble mean. This is imposed by setting

$$B(t) = B_0 \exp\left[-\frac{(t-1945)^2}{2B_d^2}\right],$$ (6)

where $B_0$ is the magnitude of the 1940s pycnocline displacement (away from the internal variability component) in metres and $B_d$ is its duration (figure 2c). $B_0$ and $B_d$ are variable parameters, whose posterior distributions are inferred using the CES procedure, i.e. our parameter inference provides estimates of how long and how large the 1940s event was in terms of its effect on the pycnocline position.

The final component of the pycnocline position (5) is the anthropogenic trend in forcing,

$$T(t) = T_0 \left(\frac{t-1960}{100}\right) \times H(t-1960),$$ (7)





where $T_0$ is the per-century-trend in pycnocline height in metres, and $H(\xi)$ is the Heaviside function, taking the value of one for $\xi \geq 0$ and zero for $\xi < 0$ (in particular, this means that the anthropogenic trend in forcing continues indefinitely). $T_0$ is considered a variable parameter, to be inferred from the CES procedure; in making inference into this parameter, we are addressing the question: how large does the anthropogenic trend in forcing have to have been to explain the PIG retreat that has occurred over the industrial era? Note that in setting the trend according to (7), we assume that the trend begins in the 1960s and is linear, which is consistent with trends in shelf-break winds (Holland et al., 2019, 2022).

## 2.2 Observationally-constrained posterior parameter distributions using CES

We aim to produce observationally constrained posterior distributions of the set of model and climate parameters, $\Theta = (A_0, C_0, M_0, B_0, B_d, T_0)$. To do so, we follow the calibrate, emulate, sample procedure outlined in Cleary et al. (2021). In this section, we describe each of these steps in detail, as well as the observations used as constraints. For each of the 14 realizations of internal variability in the forcing from the autoregressive process, we produce a unique posterior distribution of the parameters. Samples from these posterior distributions are then taken and run over the industrial period (see section 3) to reconstruct the behaviour of PIG since 1750 and facilitate an attribution assessment.

The steps of the CES procedure are as follows: in the first step (calibrate), an adaptive procedure (here, EKI) is used to obtain parameter samples that minimise model–observation error. Having many samples in the 'likely' region of parameter space, where the model–observation error is small, ensures that the emulator constructed in the following step (emulate) is accurate (with low uncertainty) for parameters in this region. This emulator gives statistical approximations to the model output, and thus model–observation error, everywhere in parameter space. In the final step (sample), Markov-chain Monte Carlo sampling is used to generate posterior distributions of model and climate parameters, with step updates in the Markov-chain based on the approximate model–observation error constructed in the emulate step.

### 2.2.1 Prior distributions of model and climate parameters

Prior distributions of model parameters (i.e. before observations have been assimilated) are assumed to be all Gaussian, with mean and standard deviation as shown in Table 1. The prior mean on the basal melt rate exponent prefactor ($M_0 = 0$) results in a melt rate parametrization $\dot{m} = 5.0T_*^2$; as mentioned, this choice ensures that the basal melt flux in the present day with warm ambient conditions ($P_c = -400$ m) matches the basal melt flux with corresponding conditions in 2009, the warmest year on record (Dutrieux et al., 2014). In lieu of further information, we assume broad priors on model and climate forcing parameters. Crucially, we assume that the prior mean on the anthropogenic trend in forcing is zero, so as not to precondition the procedure to select an anthropogenic trend.

### 2.2.2 Observations used in the CES procedure

Posterior distributions of model and climate parameters are determined by assimilating observations of ice sheet behaviour. Here, we use observations of grounding line position in 1930 and 2015, and the volume of grounded ice in 2015. The grounding





| Parameter name | Description | Units | Prior mean | Prior standard deviation |
|:---:|:---:|:---:|:---:|:---:|
| $A_0$ | Ice viscosity prefactor | - | 1.0 | 0.3 |
| $C_0$ | Basal sliding prefactor | - | 1.0 | 0.3 |
| $M_0$ | Melt rate exponent prefactor | - | 0.0 | 1.2 |
| $B_0$ | 1940s event magnitude | metres | 200.0 | 100.0 |
| $B_d$ | 1940s event half duration | years | 5.0 | 2.5 |
| $T_0$ | Anthropogenic trend since 1960s | metres | 0.0 | 200.0 |

**Table 1.** Descriptions of the model and climate parameters and values of their prior means and standard deviations.

line position is measured along the centreline indicated in figure 1, and is determined as the weighted average of the positions of the first partially grounded cell and the last full floating cell, with the grounded fraction of a grid cell determined using the procedure outlined in Seroussi and Morlighem (2018). The 1930 grounding line position is taken as that from the initial state geometry, i.e. before any retreat has been initiated. The 2015 grounding line position and grounded volume are computed from the model ice sheet state following the inversion with present-day observations. In the following, we express the grounding line position as a grounding line retreat measured relative to the 1930 grounding line position. We assume that both observations of grounding line position are relatively poorly constrained, taking the observational errors in these quantities to be 3 km, equivalent to one grid cell. The observational error on the 2015 grounded volume is taken as 1% of this value. For simplicity, we assume that the observations are uncorrelated.

### 2.2.3 Calibrate

In the calibrate step of the procedure, we use an ensemble Kalman inversion, an iterative procedure in which model parameters are updated so that model outputs are nudged progressively closer to observations. In simple terms, there are four main substeps within the calibrate step: firstly, an ensemble of parameter samples are chosen and, for each, the model is run and we calculate how close its prediction is to the real observations. Then, we calculate the overall pattern of how far off the parameter samples from the observations are, which informs the direction in which parameter samples need to move. Next, based on the average error and how sensitive the model is to changes in the parameters, the parameters are nudged, with each guess moving in the direction that reduces the error. Finally, the model is run again with these updated guesses and the update step is repeated until the model predictions are close enough to the observations.

In more detail, the EKI is initialized by sampling $J$ times from the prior distribution of model parameters, giving a set of parameter values $\Theta_0^1, \ldots, \Theta_0^J$, where the '0' subscript refers to this being the zeroth iteration (in this section, we use subscripts to denote the iteration number and superscripts to denote the samples within this iteration).

After the $i^{\text{th}}$ iteration, the $j^{\text{th}}$ sample is updated with an update step (second term on the right hand side below) to (Cleary et al., 2021; Bach and Dunbar, 2023)

$$\Theta_{i+1}^j = \Theta_i^j + C_i^{\theta\mathcal{G}} \left( \Gamma + C_i^{\mathcal{G}\mathcal{G}} \right)^{-1} \left[ y - \mathcal{G} \left( \theta_i^j \right) \right]. \tag{8}$$



**Figure 3.** (a)–(b) Trajectories of (a) grounded volume and (b) grounding line retreat in simulations of Pine Island Glacier. Colours correspond to the different iterations, as indicated by the colour bar in (a). Results are shown for a single realization of forcing. Red points indicate observational constraints used on trajectories in the EKI, with error bars indicating uncertainties in observations (note that the error bar does not extend beyond the point in (a)). (c)–(h) Scatter plots of grounded ice volume (grv) in 2015 as a function of (c)–(e) model and (f)–(h) climate parameters for simulations whose trajectories are shown in (a)–(b). Colours of points correspond to iterations as indicated in (a). In each panel, the red dashed line indicates the observational constraint. (i) Mean (over ensemble members) absolute error in the grounded volume as a function of iteration number. Results are shown for all 14 realizations of forcing. (j) Histograms of absolute errors in the grounded volume obtained from the EKI (red) and a Latin hypercube sampling method (blue), for a single realization of forcing. In both cases, the ensemble consists of 100 simulations.





The update step is expressed as the product of a step size, $C_i^{\theta \mathcal{G}} \left( \Gamma + C_i^{\mathcal{G}\mathcal{G}} \right)^{-1}$, and a model-observation error, $y - \mathcal{G}\left(\theta_i^j\right)$. In the model–observation error term, $y$ are the observations (i.e. an array containing the grounding line position in 1930 and 2015, and the ice volume in 2015) and $\Gamma$ is the error in the these observations. $\mathcal{G}\left(\Theta_i^j\right)$ are the modelled values corresponding to the observations, i.e. the simulator $\mathcal{G}$ evaluated at parameter values $\Theta_i^j$.

In the step size, we have

$$C_i^{\Theta \mathcal{G}} = \frac{1}{J} \sum_{j=1}^{n} \left[ \left( \Theta_i^j - \bar{\Theta}_i \right) \left( \mathcal{G}(\Theta_i^j) - \bar{\mathcal{G}}_i(\Theta) \right)^{\mathrm{T}} \right], \tag{9}$$

which plays the role of a cross-covariance between parameters $\Theta$ and model outputs $\mathcal{G}(\Theta)$, encoding how changes in the model parameters relate to changes in the model predictions. Here, the overbar denotes an average over the ensemble members at this iteration. In addition,

$$C_i^{\mathcal{G}\mathcal{G}} = \frac{1}{J} \sum_{j=1}^{J} \left[ \left( \mathcal{G}(\Theta_i^j) - \bar{\mathcal{G}}_i(\Theta) \right) \left( \mathcal{G}(\Theta_i^j) - \bar{\mathcal{G}}_i(\Theta) \right)^{\mathrm{T}} \right] \tag{10}$$

plays the role of the output covariance of $\mathcal{G}(\Theta)$, encoding how much the model outputs vary together across the ensemble. Together with parameters-output covariance (sensitivity of model outputs to parameters) and the error in the observations, the output covariance sensitivity controls how big the update step should be.

In the results generated in this study, we use $J = 20$ ensemble members and five iterations, giving 100 simulations for each realization of forcing. The total number of simulations is thus $1400 = 20$ ensemble members $\times 5$ iterations $\times 14$ realizations of forcing. There is no rule on how many ensemble members and iterations should be used. However, we found that with fewer ensemble members ($J = 10$), parameter updates (8) would be swamped by some ensemble members with large model–observation errors, which would drive the whole ensemble away from the optimal parameter values. Other studies (e.g. Cleary et al., 2021; Mansfield and Sheshadri, 2022), with different model operators $\mathcal{G}$, have found that smaller $J$ yield good convergence.

Figure 3a–b demonstrate how trajectories of grounded ice volume and grounding line retreat converge as iterations proceed, indicating the potential of the EKI to obtain trajectories of ice sheet change which are consistent with observed changes. Corresponding parameter values are shown in figure 3c–h. Parameter values do not converge to a single value, but towards a distribution, which are approximate samples from the posterior of the parameters.

We find that in each realization of forcing, by the fifth iteration the mean absolute error – the mean (over the ensemble members) of the model–observation error – in the grounded volume is below 1% (figure 3i).

It is interesting to compare the EKI procedure with Latin hypercube sampling (figure 3j), the current standard sampling method in ice sheet modelling. Latin hypercube sampling ensures that the parameter space is evenly covered, usually by applying a space-filling algorithm which ensures that the distance between points is maximised. To generate a Latin hypercube sample, a fixed range of parameters must be specified; here, we set these limits to be two standard deviations either side of the mean of the prior distributions. Histograms of errors in the grounded volume (figure 3j) confirm that the errors in the EKI ensemble (based on all iterations of the EKI) are significantly smaller than those in the LHC ensemble: for this particular





problem, EKI sampling is superior to LHC sampling when assessed in terms of observational constraints. Additionally, errors in the final iteration of the EKI are very small (left most bin of figure 3j) – much more accurate than in the LHC.

### 320  2.2.4  Emulate

In the next step, we train emulators–computationally cheap statistical approximation of the expensive simulator–on the calibrated simulations described in the previous section. Essentially, the emulators are cheap models, based on the expensive simulations, which approximate the model's prediction of the three observational values. Doing so allows us to evaluate the model output, and thus construct model–observation errors, for any parameter values (i.e. not just those sampled with model

simulations during the calibration step). This model–observation error is used in the Markov chain Monte Carlo (MCMC) sampling, described in the following section.

In more detail, the simulator can be considered a map $M_s$ between input parameters $\Theta$ and predictions of the quantities for which we have observations:

$$\text{(1930 grounding line, 2015 grounding line, 2015 grounded ice volume)} = \mathbf{M}_s(\Theta). \tag{11}$$

The emulators are approximations to the map $M_s$. We construct emulators for each of the observational constraints individually, giving maps $M_e^1(\Theta), M_e^2(\Theta), M_e^3(\Theta)$ where

$$\text{(1930 grounding line, 2015 grounding line, 2015 grounded ice volume)} = \mathbf{M}_s(\Theta) \approx (M_e^1(\Theta), M_e^2(\Theta), M_e^3(\Theta)). \tag{12}$$

Importantly, the functions $M_e^1(\Theta), M_e^2(\Theta), M_e^3(\Theta)$ can be evaluted *everywhere* in parameter space, whereas the map $\mathbf{M}_s(\Theta)$ is only known at the (100) discrete values where the simulations exist. The emulators match the simulated values for each

point in the parameter space where there is a simulation, and provide predictions for all other points in the parameter space, alongside an estimate of uncertainty in this prediction.

This process is repeated for each realization of forcing, giving $14 \times 3 = 42$ emulators in total. Each emulator is a Gaussian process–a non-parametric method that treats the simulator as an unknown function of its inputs (O'Hagan, 2006). Importantly, Gaussian processes return analytic uncertainty estimates, which are not directly available in many other emulation methods,

such as neural networks (Gawlikowski et al., 2023). These analytic uncertainty estimates are propagated through the MCMC sampling to ensure that emulator error is robustly represented in posterior distributions of model and climate parameters. We train the emulators on all of the simulations from the EKI, rather than just those from the final iteration as suggested by (Cleary et al., 2021), as this was found to improve emulator performance.

To construct the emulators, we use the R package RobustGaSP (Gu et al., 2019). Each emulator uses a Matérn 3/2 kernel,

which we found displayed the best coverage (the percentage of emulator predictions falling outside two emulator standard deviations), and lowest RMSE of five standard kernel choices: Matérn (3/2), Matérn (5/2), and three members of the power exponential family with high, medium and low exponent values ($\alpha = 1.9$, $\alpha = 1.0$, and $\alpha = 0.1$) in a leave-one-out-cross-validation experiment (Bastos and O'Hagan, 2009), described in the next paragraph.

To validate our emulators, we perform a leave-one-out-cross-validation. In this procedure, each emulator is sequentially

trained on the output of every set of parameter values used in the EKI except for one (i.e. on 99 parameter sets), and then



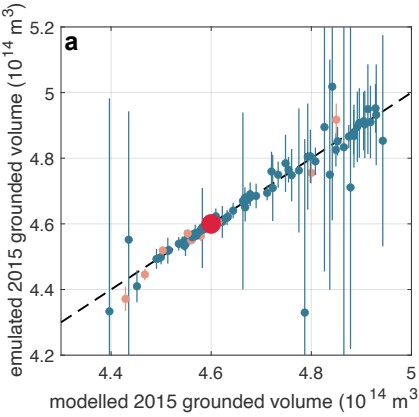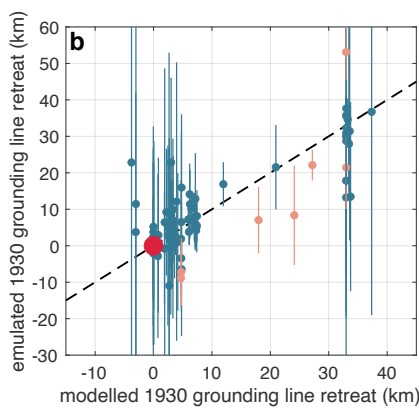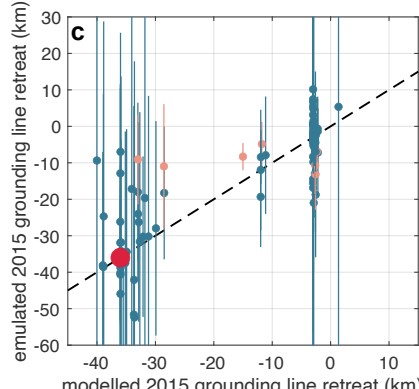

**Figure 4.** Results of a leave-one-out-cross validation of model emulators. Each panel contains a scatter plot of modelled versus emulated quantities, as follows: (a) 2015 grounded ice volume, (b) 1930 grounding line retreat relative to the pre-industrial position and (c) 2015 grounding line retreat relative to the pre-industrial position. In each case, blue (orange, respectively) markers are used for simulations in which the emulated quantity is inside (outside) one emulator standard deviation of the modelled value. Error bars indicate the standard deviation of each emulator prediction. Black dashed lines are 1-1 lines, onto which each point would fall if the emulated values perfectly matched the modelled values. Results are shown for a single realization of forcing.

evaluated on the remaining parameter set. For each, we note the emulator error and uncertainty in the prediction. We use the coverage as a metric for emulator performance (figure 4a). We find that most emulators have coverages between 80% and 90% (mean 85%, 87%, and 91% standard deviation 3%, 4%, and 3% for 2015 grounded ice volume, 1930 grounding line retreat and 2015 grounding line retreat, respectively), reflecting good performance. In addition, many of the data points outside the

two emulator standard deviation bands have low root mean squared error.

### 2.2.5 Sample

Having emulated model predictions of observational constraints, we construct posterior distributions of model and climate parameters using MCMC sampling of the emulators. The sampling step uses the fast emulator to explore a wide range of possible inputs quickly, generating many outputs. This allows us to determine which parameter values are likely (and which

aren't) based on agreement between outputs and observations, without having to run the slow, full simulation repeatedly.

The Markov chains move through the space of model and climate parameters $\Theta$. Each is initiated using parameter values taken as the mean of the final iteration of the EKI: writing $\Phi_n$ for the state of the Markov Chain at its $n^{\text{th}}$ step, we take $\Phi_0 = \bar{\Theta}_5$. Following Cleary et al. (2021), we use a Metropolis-Hastings algorithm for the update step of the Markov chain. The size of this update step in this algorithm is determined from a multivariate Gaussian distribution whose covariance given by the empirical

covariance of the ensemble from the EKI – essentially the spread and relationship of the previous samples from the EKI (see below). Our choice of initial state and update step size pre-condition the Markov chain with approximate information about the



posterior distribution of model and climate parameters from the EKI, allowing the MCMC to be initialized into a state close to the posterior mean.

Explicitly, the MCMC algorithm is as follows (see italic text at the start of each step for a non-mathematical explanation):

1. *Start the procedure with the parameters corresponding to the end of the EKI procedure.*

   Initiate the MCMC with parameter values $\Phi_0 = \bar{\Theta}_5$.

2. *To make a step, suggest new values of the parameters, which are a step away from the current values. The size of this step depends on the spread and relationship of the samples from the EKI.*

   At the $k^{\text{th}}$ step in the chain, propose a new Markov chain state $\Phi_{k+1}^* = \Phi_k + \zeta_k$ where

$$\zeta_k \sim \mathcal{N}\left(0, \frac{1}{J}\sum_{j=1}^{J}\left[(\Theta_5^J - \bar{\Theta}) \otimes (\Theta_5^J - \bar{\Theta})\right]\right), \tag{13}$$

   with $\mathcal{N}$ denoting a normal distribution and $\otimes$ is the tensor product.

3. *Calculate the relative likelihood of the previous and proposed parameters. This likelihood is based on how well the emulator outputs for both previous and proposed parameters agree with observations, as well as the prior likelihood of these parameters (before any data assimilation) and how uncertain the emulator is for these parameters. Lower*

*likelihoods are obtained for parameter combinations for which emulator-observation error is higher, for which the prior is less likely and/or for which the emulator is more uncertain.*

   Compute the acceptance probability $a(\Phi_k, \Phi_{k+1}^*)$ where

$$a(s,t) = \min\left\{1, \exp\left[\left(\Psi^M(s) + \frac{1}{2}||s||_{\Gamma_\theta}^2\right) - \left(\Psi^M(t) + \frac{1}{2}||t||_{\Gamma_\theta}^2\right)\right]\right\} \tag{14}$$

   where

$$\Psi^M(\Theta) = \frac{1}{2}\left(||y - \mathcal{G}(\Theta)||_{\Gamma_{GP}+\Gamma_y}\right)^2 + \frac{1}{2}\log\left\{\det\left[\Gamma_{GP}(\Theta) + \Gamma_y\right]\right\}, \tag{15}$$

   with $||a||_A = ||A^{-1/2}a||$ for any positive-definite matrix $A$ and $||.||$ denoting the L2-norm. $\Gamma_{GP}(\Theta)$ and $\Gamma_y$ are the covariance matrices of the emulators and the observations, respectively, and $\Gamma_\theta$ is the covariance matrix of the prior distribution of model and climate forcing parameters $\theta$.

4. *Decide whether to move to the proposed parameters based on the likelihood calculated in the previous step.*

   Update the chain to $\Phi_{k+1}$ to $\Phi_{k+1}^*$ with probability $a(\Phi_k, \Phi_{k+1}^*)$; otherwise set $\Phi_{k+1} = \Phi_k$.

5. *Repeat the previous steps many times.*

   Repeat steps 2–4 $S$ times. In the results presented here, we use $S = 50000$.

6. *Remove some samples from the start, where the chain hasn't yet settled into the right area.*

   Remove the first $S_{\text{burn}}$ samples from the sequence (the so-called 'burn in' period). In the results presented here, we use
   $S_{\text{burn}} = 1000$.



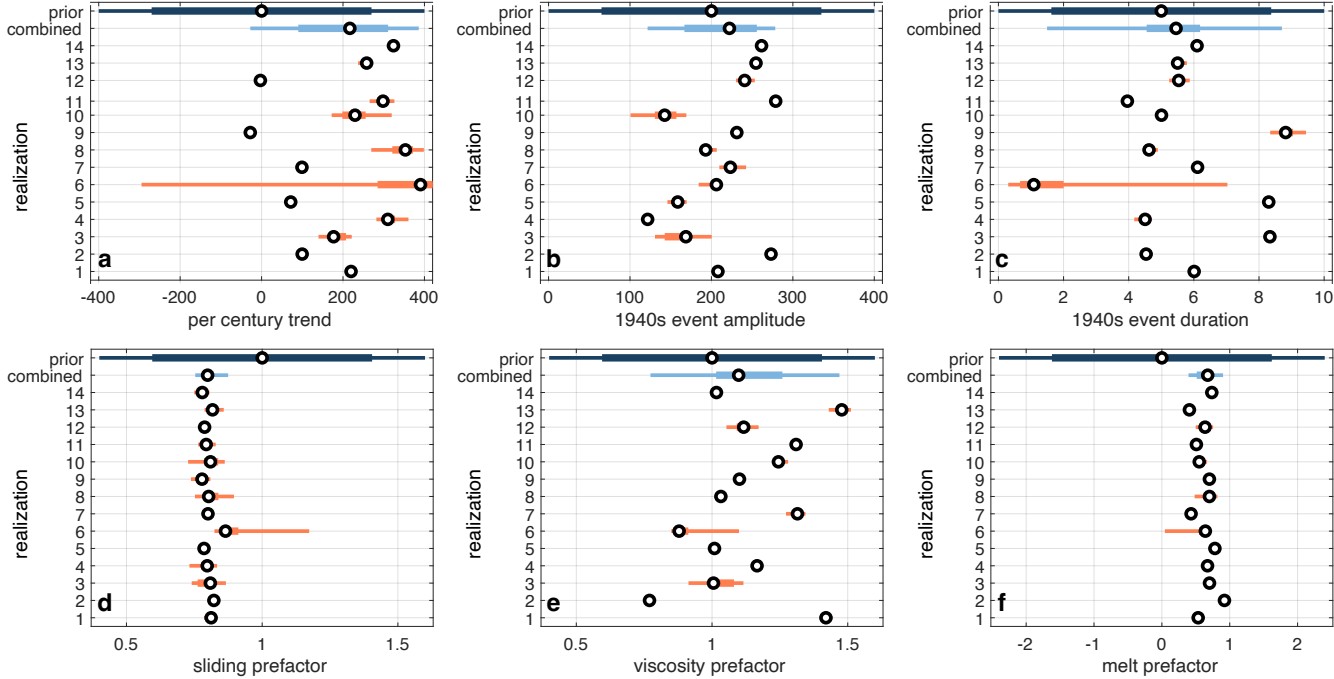

**Figure 5.** Posterior distributions of model and climate parameters. Each panel corresponds to an individual parameter as labelled on the abscissae. Distributions are shown for each realization of forcing (numbered 1–14, orange), for the combined distribution (light blue), and for the prior (dark blue). For each, narrow lines indicate the 95% interval, thick lines indicate the interquartile range, and black points indicate the median.

Samples from the MCMC yield posterior distributions of model and climate parameters. Note that the likelihood (15) consists of a first term corresponding to the model–observation error and a second term corresponding to uncertainty in both the observations $\Gamma_y$ and the emulator $\Gamma_{GP}(\Theta)$, i.e. emulator uncertainty is explicitly accounted for in the posterior distributions of model and climate parameters.

**3 Results**

**3.1 Posterior distributions of climate and model parameters**

Posterior distributions of climate and model parameters are shown in Figure 5. In general, the posterior distributions are much tighter than the priors, indicating that the three observational constraints assimilated during the CES procedure provide strong controls on model and climate forcing parameters. In other words, the region of parameter space which is compatible with 405 the precise retreat of PIG over the 20th century is small, despite our choice to permit fairly broad errors on observational constraints. Although the posterior distributions for each realization of forcing are relatively narrow, the combined posterior





distribution (obtained by taking all samples from the 14 individual CES procedures) can remain broad, particularly for the ice viscosity prefactor (Figure 5b), anthropogenic trend (Figure 5d), and the 1940s event duration (Figure 5f). This indicates that aleatoric uncertainty is a major contributor to uncertainties in model and climate parameters. This is consistent with previous

work (Robel et al., 2019), demonstrating that aleatoric uncertainty plays an important role in ice-ocean systems, particularly those which are subject to strong ice-ocean feedbacks. Thus, we re-iterate that ice sheet modelling studies should consider multiple realizations of forcing when making assessments of future and past change. Furthermore, internal variability in the chaotic climate system is random and unpredictable, so this demonstrates an irreducible source of uncertainty inherent in all ice sheet predictions.

Our observationally constrained posterior distributions provide inference into both the past climate as well as the physical model. We find that the posterior distribution on the basal sliding prefactor is extremely tight relative to the prior, (mean: 0.79, 95% confidence interval: [0.75, 0.87]). This indicates that the field of basal sliding coefficients should be adjusted by approximately 20% to correctly capture the evolution of the Pine Island Glacier over the 20th century, within the context of the parameters varied and model structure (see below).

Similarly, the posterior distribution on the melt rate exponent prefactor has mean 0.67 (95% confidence interval: [0.39, 0.9]), indicating that the melt rate is significantly higher (by a factor of 1.59, 95% confidence interval: [1.31, 1.87]) than the prior. Recall that the prior is constrained by observations: the prior mean is determined by matching modelled and observed meltwater fluxes from Pine Island Ice Shelf. Thus, our model predicts that higher melt rates are required to drive retreat than are observed.

These shifts, towards lower basal friction and higher basal melt rate, suggest that the CES procedure is shifting parameters to make the model more sensitive, either because other feedbacks which increase ice sheet sensitivity (e.g. De Rydt et al., 2014; De Rydt and Naughten, 2024; Bradley et al., 2022; Bett et al., 2020; Bradley and Hewitt, 2024; Holland et al., 2023) are not included in our model, or because the model structure (e.g. basal sliding parametrisation, melt rate parametrization) is not correct.

Our simulations suggest that the 1940s event was a 222 m (95% confidence interval: [121, 278] m) shoaling of the pycnocline (note that this must be superimposed on the internal variability which, although close to zero, is non-zero because of the finite number of realizations of forcing, see figure 2). The inferred median duration of this event was 5.45 years (95% confidence interval: [1.49, 8.70] years). The shallowness of the implied pycnocline depth perturbation would represent an extremely large ocean anomaly, warmer than any event in the observational record (Dutrieux et al., 2014) and comparable to modelled

conditions for the year 2100 (Naughten et al., 2023). However we know that this climatic event must have been exceptional, as it stands out in regional climate proxies (Schneider and Steig, 2008; O'Connor et al., 2023) and induced substantial, widespread ice sheet change (Smith et al., 2017; Clark et al., 2024). This demonstrates the power of the CES procedure not only to infer information about model parameters, but also about the climate forcing.

Importantly, our observationally constrained posterior distribution of anthropogenic trend reveals that it is unlikely that

the 20th century PIG retreat would have occurred without anthropogenic forcing. The posterior mean suggests a 216 m / century (95% confidence interval: [-27, 386] m) shoaling of the pycnocline is required to reconstruct the observed retreat.





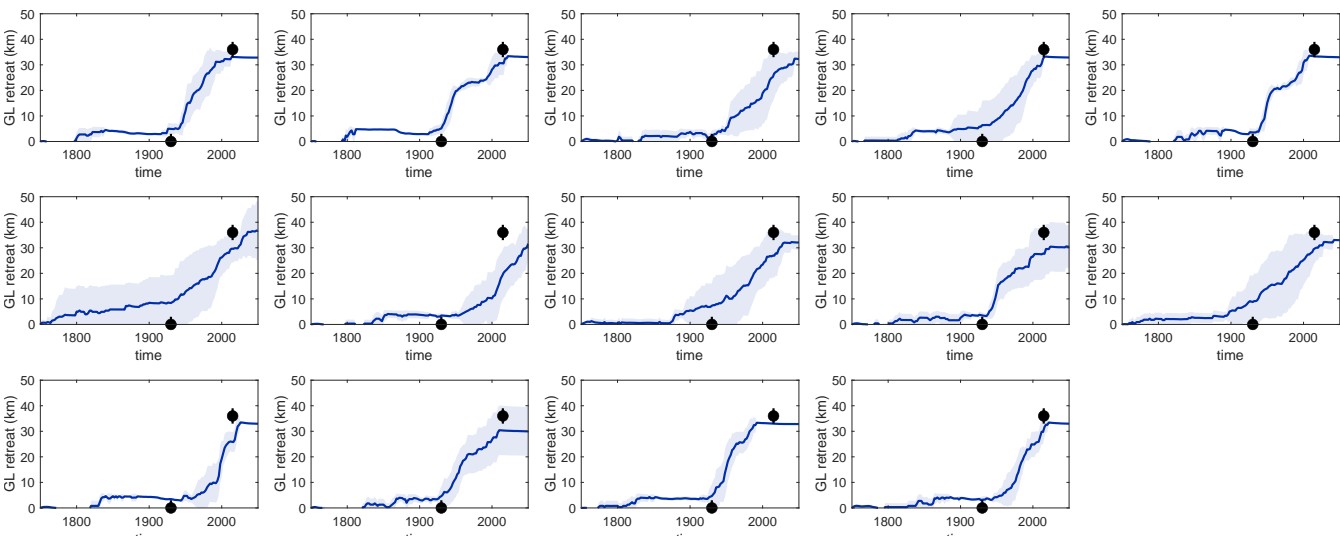

**Figure 6.** Trajectories of grounding line retreat from posterior samples. Each panel corresponds to an individual realization of forcing. The shaded band indicates one standard deviation around the central estimate (solid curve). Black points with error bars indicate observational constraints.

This would imply an approximately 100 m pycnocline shoaling between 1960 and 2010, which is consistent with ocean model simulations (Naughten et al., 2022, 2023). A student T-test reveals that the posterior mean anthropogenic trends over the realizations of forcing are significantly different from zero ($p < 0.001$). The probability that the combined posterior distribution

is negative (i.e. that the trend was zero or negative) is 16%. In the following section, we explore the implications of these likely non-zero trends in forcing on the PIG retreat that took place over the 20th century.

### 3.2 Past and future retreat of Pine Island Glacier

Having constructed posterior distributions of model and climate parameters, we reconstruct the statistics of past, and possible future, retreat of the Pine Island Glacier by sampling from these parameter distributions and using these to perform a set of

optimised model simulations. For each realization of forcing, we take 10 samples from the posterior distributions (giving 140 simulations in total), and run these from 1750, with the same configuration as in the calibration step. We run these simulations out to 2300, rather than 2050 as above (note these samples are not necessarily in the initial EKI ensembles). (Note that it would also be possible to obtain the statistics of retreat trajectories by emulating time-series of model evolution obtained during the calibration step; however, any evolution beyond 2050 would be outside the training set of such an emulation method, potentially

introducing inaccurate results.)

Trajectories of grounding line retreat for each realization of forcing are shown in Figure 6. These trajectories are data-constrained reconstructions of grounding line retreat, with quantified uncertainty arising from incomplete knowledge of model and climate forcing parameters. All but one of our reconstructed grounding line trajectories (and grounded volume trajectories,




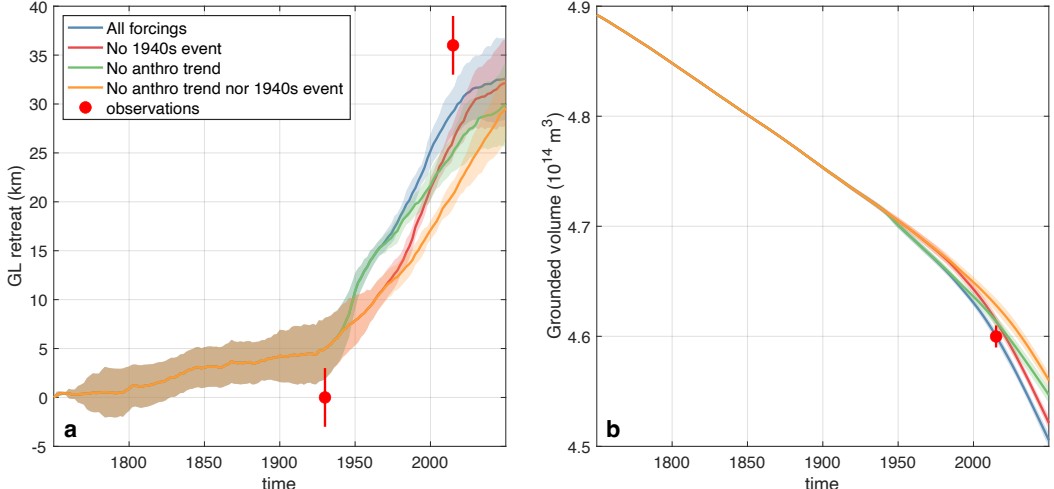

**Figure 7.** Trajectories of (a) grounding line retreat and (b) grounded volume from posterior samples. Results are shown for several different ensembles as follows: all forcings (blue), with no 1940s event (red), no anthropogenic trend (green), and neither a 1940s event nor an anthropogenic trend (orange; see main text for an explanation of these scenarios). Shaded regions indicate one standard deviation around the central estimate (solid line). Note that the trajectories and shaded regions are indistinguishable for the four scenarios prior to the 1940s. Red points with error bars indicate observational constraints.

though this is not shown) passes through the observational constraints, to within the observational and parametric uncertainty.
This confirms that the posterior distributions of model and climate forcing parameters are consistent with the observational constraints. In general, reconstructed grounding line trajectories slightly underestimate the level of retreat over the 20th century (Figure 6). We attribute this to the influence of the prior parameter distributions that we have chosen, which seem to constrain the results to have lower retreat than observed. In particular, assimilating observations of retreat nudges the distributions of model and climate parameters to have (i) lower basal sliding prefactors (figure 5a), (ii) higher basal melt rates (figure 5c), and
(iii) higher anthropogenic trends (figure 5d) than the priors, each of which promotes enhanced retreat. (The prior influences the posterior distributions via the MCMC step (14).)

In Figure 7, we show trajectories of grounding line retreat and grounded volume from all 140 samples from the posterior distributions. These trajectories (blue curves in Figure 7) represent our best estimates of reconstructed grounding line position and ice volume, accounting for both aleatoric and parametric uncertainty. As for the trajectories for individual realizations
of forcing, the posterior samples reproduce the observed retreat, albeit with a slight underestimation. Uncertainties in these trajectories are relatively small, indicating that, as for the posterior distributions, even our small number of observational constraints provides strong constraints on trajectories.

In the previous section, we demonstrated that it is unlikely that the observed retreat of Pine Island Glacier over the 20th century would have occurred without an anthropogenic trend in forcing, and that it is likely there was a sustained anomaly in
forcing in the 1940s. We now move on to assess how much additional grounding line retreat and ice loss took place because of




this likely trend and 1940s event. To do so, we construct three further ensembles of simulations, supplementing the posterior samples described above (which is referred to henceforth as 'all-forcings' ensemble). Each of these further ensembles are identical to the 'all-forcings' ensemble, with 140 simulations, but with the following climate forcing components changed: in the first, all samples have no anthropogenic trend in forcing, i.e. $T_0 = 0$ (referred to as the 'no-trend' ensemble); in the second,

all samples have no 1940s event, imposed by setting $B_0 = 0$ (referred to as 'no 1940s event' ensemble); in the third, all samples have no anthropogenic trend in forcing nor 1940s event ($T_0 = 0,\ B_0 = 0$, referred to as 'no trend nor 1940s event' ensemble).

Trajectories of grounding line retreat and grounded volume for these ensembles are shown in figure 7. Prior to the 1930s, each of the four ensembles have identical trajectories. Shortly after the 1940s, trajectories in the 'all-forcings' and 'no-trend' ensembles display rapid grounding line retreat, indicating that our reconstructions capture the role of the 1940s event as a

trigger for the present phase of retreat. However, even in the 'no 1940s event' and 'no trend nor 1940s event' ensembles, grounding line retreat still accelerates after the 1940s (red and orange curves in figure 7), despite the anthropogenic trend in forcing only entering from 1960 (where present). We discuss possible reasons for this in section 4. By the 1970s, the effect of the anthropogenic trend starts to emerge: the all forcings (blue) and no 1940s event (red) ensembles display enhanced grounding line retreat rates compared to the no trend (green) and no trend nor 1940s event (orange) ensembles, respectively.

To quantify the role of the anthropogenic trend and 1940s event in the retreat of PIG, we compare the magnitude of retreat that has taken place in these ensembles by 2015. We focus solely on grounding line retreat, rather than grounded volume, because this metric has a 'before' observation – the grounding line position prior to the 1940s – and an 'after' observation – the grounding line position in 2015 (the ice volume has no such before constraint). We calculate the fraction of attributable retreat as

$$\text{fraction of attributable retreat} = \frac{\text{all forcings retreat between 1930 and 2015} - \text{no trend retreat retreat between 1930 and 2015}}{\text{all forcings retreat retreat between 1930 and 2015}}.$$

495                                                                                                                                              (16)

The fraction of attributable retreat is interpreted as the fraction of the retreat which the anthropogenic trend in forcing is responsible for. A value of 1, for example, would indicate that anthropogenic forcing is responsible for 100% of the retreat. Our simulations indicate that the fraction of attributable retreat is 0.18. In other words, anthropogenic trends in forcing enhanced the retreat of PIG since the 1940s and are responsible for just under one fifth of the retreat over the industrial era. This is

equivalent to an excess grounding line retreat of 4.3 km.

Repeating the calculation (16) for the 1940s event (i.e. replacing the no trend retreat with no 1940s event retreat in (16)) reveals that 1940s event is responsible for around 13% of the grounding line retreat of PIG over the industrial era, corresponding to an excess grounding line retreat of 3.2 km.

Figure 7a indicates that the anthropogenic signal in retreat has only just started to emerge (the green and blue lines have

only recently deviated from one another). This begs the question of whether the anthropogenic signal of retreat will continue to persist beyond climatic noise. To address this question, we consider the evolution of the ensembles beyond the present day. To produce these future continuations, we extend the generation of autoregressive internal variability for each of the 14 realisations, and simply extend the pycnocline trends forward in time, where present, so that the pycnocline rises indefinitely.



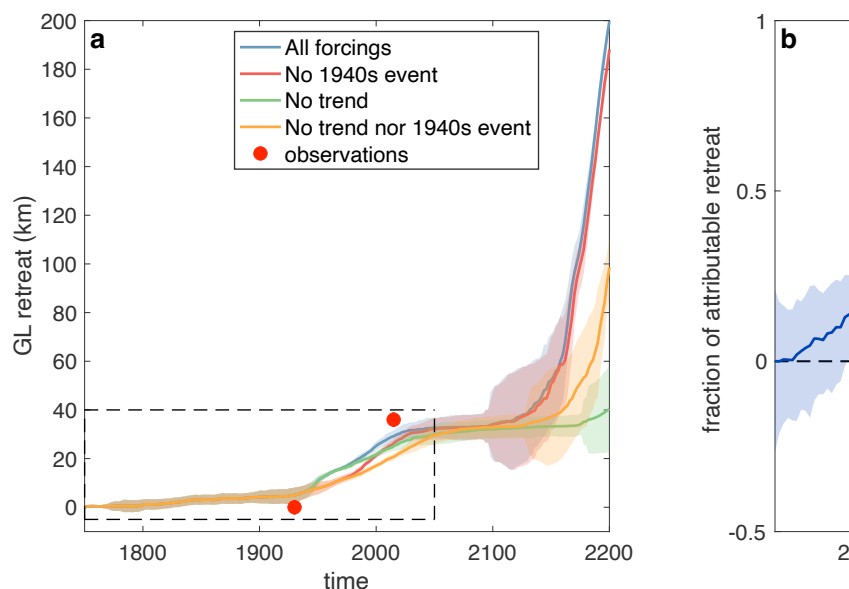
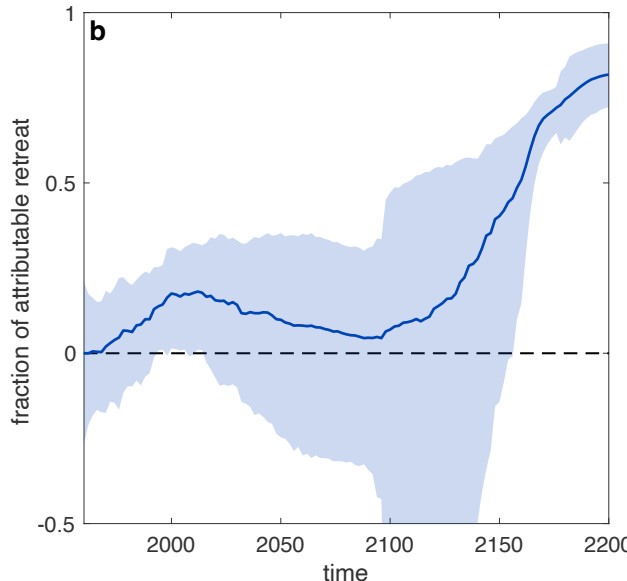

**Figure 8.** (a) As in figure 7a, but for simulations extended out until 2200. The dashed box indicates the area shown in figure 7a. (b) Time evolution of the fraction of attributable retreat (equation (16)). The solid line shows the central estimate, obtained using the mean of trajectories shown in (a). The filled area shows error in this quantity.

In these simulations (Figure 8a), the retreat of all ensembles temporarily stabilizes at a location roughly equivalent to the present day grounding line position, by the mid-end of the 21st century. This position corresponds to a local topographic high (figure 1c), and is consistent with locations of relative PIG stability (Rosier et al., 2021; Reese et al., 2023). The all forcings and no 1940s event ensembles reach this position synchronously, at around 2050; by this time, approximately 100 years after it occurred, simulations have lost memory of the 1940s event. This highlights, however, the long timescales on which even relatively short periods of anomalously high melting can influence glacial retreat. Between approximately 2075 and 2100, all four of the ensembles are grounded at or close to this local topographic high; during this time, the fraction of attributable retreat approaches zero (calculated by replacing the 2015 in equation (16) with the relevant year, figure 8b), i.e. the anthropogenic signal is undetectable in the grounding line position.

Around 2100, retreat is reinitiated in the central estimate of ensembles with an anthropogenic trend, while those without a trend remain at the topographic high for longer. Concomitantly, the fraction of attributable retreat increases (note that the ensemble spread increases beyond this because several ensemble members re-advance at this point). By 2150, the fraction of attributable retreat becomes significantly different from zero. Thus, our simulations suggest that while the anthropogenic signal in the retreat is detectable at present in PIG's grounding line position, this will not persist beyond the 2030s, and the next time that this signal will be present is in the middle of the 22nd century. By 2200, almost all of the retreat is attributable to anthropogenic trends in forcing.



We also note that deglaciation occurs sooner in the ensemble with no trend and no 1940s event, compared to that with just no trend. We suspect is because of the role of history in the retreat (as we have elaborated above): the retreat in the no trend no event ensemble over the 1950-2050 period occurs more steadily, which we suggest preconditions it for sooner retreat off the temporarily stable ridge.

## 4   Discussion

In this paper, we have made progress in two areas: firstly, we have demonstrated how the calibrate-emulate-sample framework (Cleary et al., 2021) may be used to efficiently infer observationally-constrained posterior distributions of model and climate forcing parameters in ice sheet models; and, secondly, we have used the results of this to perform an attribution assessment on the retreat of the PIG over the industrial era and into the future. Our simulations suggest that it is unlikely that the magnitude of industrial era PIG retreat would have occurred without an anthropogenic trend in forcing (it is unlikely that the

anthropogenic trend in forcing is negative) and that the likely trends in forcing that occurred since the 1960s enhanced the retreat of PIG. We additionally quantified this effect, finding that anthropogenic trends in forcing since the 1960s are responsible for approximately 20% of the PIG retreat over the industrial era. Our study is, to the best of our knowledge, the first to quantify the role of anthropogenic trends in forcing on retreat of an individual glacier in an ice sheet (i.e. not a mountain glacier). This study builds upon frameworks for attributing retreat of such glaciers (Christian et al., 2022; Bradley et al., 2024b) and supports

the growing literature on attribution of mountain glacier retreat (Vargo et al., 2020; Stuart-Smith et al., 2021; Marzeion et al., 2014; Roe et al., 2021) demonstrating that anthropogenic forcing has a strong influence on glacier evolution.

    Our analysis also indicates that the 1940s event, identified as being synchronous with the beginning of the phase of rapid retreat (Smith et al., 2017), also played an important role in the retreat of PIG over the industrial era. However, the ice sheet still retreats significantly in our simulations with no anthropogenic trends in forcing nor 1940s event. We suggest two possible

reasons for this: one related to the physical system and one related to our experimental setup. Regarding the experimental setup, we suggest that our choice of initial state and pre-industrial forcing may be important. The initial state is obtained by advancing the grounding line to the seabed ridge and then applying a cold forcing spin-up. This forcing, corresponding to the coldest conditions obeserved in recent years (Webber et al., 2017), means that the ice sheet at the start of our simulations is likely larger than it would have been in practice. This can be seen in trajectories of grounded volume evolution (Figures 3b

and 7b): the grounded volume must necessarily shrink in order to match the 2015 observational constraint. In addition, our choice of historical forcing is consistent with present day variability and not preindustrial values and may therefore be too strong (though it is consistent with our prior assumption on the anthropogenic trend in forcing). Forcing in the pre-1940s period of our experiments may be too strong, helping to initiate retreat even when no anthropogenic trend is imposed.

    Our choice of initial condition and historical forcing may have preconditioned the ice sheet to retreat, with the CES pro-

cedure selecting the parameters which are consistent with retreat being initiated in the 1940s, regardless of the 1940s event or anthropogenic trends in forcing after the 1960s. This experimental setup may therefore obscure the role of anthropogenic trends in forcing, meaning that the precise fraction of attributable retreat that we obtained is lower than it is in reality. To





counter this, future attribution studies of Antarctic glaciers following the procedure adopted here should either run for a longer period of time (we chose a 1750 start time as a balance between a sufficiently long pre-industrial period and computation ex-
pense), though more generally our results highlights the importance of long spin-up periods when simulating contemporaneous retreat of the Antarctic Ice Sheet. In addition, the initial state and historical forcing should be treated as additional parameters to be constrained by the CES procedure. The former could be achieved, for example, by scaling the size of the ice sheet by a prefactor (we effectively took a prior distribution on this parameter to be a delta function, with no uncertainty).

The fact that we still observe significant retreat in our simulations with no anthropogenic trends in forcing nor 1940s event
may also have a physical explanation, pointing towards an important role of long term trends in forcing – the memory of slow ice sheet retreat over the Holocene – in the retreat of the Pine Island Glacier over the 20th century. This result may suggest that the initial state is not actually a true steady state, but rather evolving slowly and reaches a critical point in the 1940s, with anthropogenic trends in forcing since the 1960s expediting this retreat. In our experiments, we included (parametrisations of) what we thought were the two main contributors to forcing: the 1940s event and the anthropogenic trend. However, the
results of the experiments suggests that we require a refocus on the quantification of the preindustrial volume and conditions, in addition to the 1940s bump and trend, and that pre-industrial ice sheet trends in forcing may also have an important role to play and should be considered in future attribution studies of the West Antarctic Ice Sheet.

We have shown that, using the CES procedure, only a small number of observational constraints can provide strong controls on permissible model parameters, resulting in narrow (relative to the priors) posterior distributions of model and climate
parameters. It is important to stress that this procedure is generic: all that is required is to construct an array of model–observation errors. Ice sheet and glacier systems are typically highly non-linear and thus the space of likely parameters may be small; for these cases, CES may be particularly effective. We demonstrated that the EKI embedded within CES outperforms latin hypercube sampling, the current state of the art in ice sheet modelling.

The procedure applied here can also be considered a transient calibration of ice sheet model parameters over a long period
(though using fewer observations and with fewer parameters in the control space – several scalars rather than whole fields – than other transient calibration studies (e.g. Goldberg et al., 2015)). Our calibrated simulations suggest that PIG will soon stabilise at a local topographic high, close to its present day grounding line position. It will remain grounded at this local topographic high for approximately 100 years, before accelerating rapidly in the middle of the 22nd century. These results assume, however, that anthropogenic trends in forcing persist indefinitely; in fact, ocean simulations suggest ocean forcing will
accelerate in the coming decades (Naughten et al., 2023), regardless of the emissions pathway followed, potentially expediting this retreat.

It is important to note that these projections implicitly assimilate (through both the initial sampling of parameters in the EKI and in the update step of the MCMC) prior distributions of model and climate forcing parameters. In this case, the prior distributions nudge the model towards lower retreat than suggested by observational constraints. Thus, future retreat may be
faster than our simulations suggest. In addition, we have used present day time-invariant ice viscosity in our simulations; in the past, however, the Pine Island Ice Shelf likely had higher structural integrity (Lhermitte et al., 2020) and thus lower effective viscosity; as the ice shelf continues to degrade in the future, its buttressing effect will reduce, expediting retreat.





Additionally, loss of ice shelf area, not included in our model, may encourage ice sheet retreat either directly via loss of buttressing (Joughin et al., 2021) or indirectly via changes in basal melt rates (Bradley et al., 2023, 2022; Yoon et al., 2022).

Projected future increases in accumulation may, however, offset these ice losses (we assume constant accumulation in this study). This assumption may also obscure attribution assessments: although reconstructed WAIS surface mass balance shows no trend since 1800, there is a negative trend over the past 1000 years (Thomas et al., 2017). These longer timescale trends might have enhanced ice sheet retreat, underscoring the role of longer term trends in the present phase of WAIS retreat. Here, we focussed on trends in ocean forcing because it is understood to be the dominant driver of recent WAIS retreat (Shepherd

et al., 2004; Pritchard et al., 2012), but future attribution assessments should also consider trends in accumulation.

Our study demonstrates that it is possible to attribute retreat of glaciers in the Antarctic Ice Sheet to anthropogenic climate change. As the World's ice sheets continue to represent an ever-increasing fraction of the global sea level rise budget (IPCC, 2022), such studies will pave the way for global attribution assessments of sea level rise, building upon those from other sources of sea level rise particularly mountain glaciers (Roe et al., 2021; Marzeion et al., 2014) and thermosteric sea level rise (Slangen

et al., 2014). Attribution studies are increasingly used to support recourse for the harms associated with anthropogenic climate change both in the courtroom (Marjanac and Patton, 2018) and in policy (Burger et al., 2020); therefore, studies attributing retreat of glaciers to anthropogenic climate change may play an important role in supporting recourse for the harms associated with sea level rise.

## 5   Conclusions

In this paper, we have used the calibrate-emulate-sample procedure to quantify the role of anthropogenic trends in forcing on the retreat of the Pine Island Glacier over the industrial era. We have demonstrated that the CES procedure is effective at constraining model and climate forcing parameters, even when only a small number of observational constraints are assimilated. We find that it is unlikely that the extent of the observed PIG retreat over the industrial era would have occurred without anthropogenic trends in forcing. In addition, anthropogenic trends in forcing almost certainly increased the industrial era

retreat of PIG. These trends are responsible for an extra 20% of the retreat over this period. Our results also suggest that the lingering effects of the long, slow retreat over the Holocene may play an important role in recent past and future retreat of the Pine Island Glacier, but our choice of initial state and pre-industrial forcing may have influenced this conclusion.

*Code and data availability.* Code to run simulations, perform the CES procedure, and make the figures in this paper are held at https://github.com/alextbradley/PIG-CES. Note that this repository contains model output for constraints (i.e. grounding line position in the 1930

and 2015, ice grounded volume in 2015); full model output, which is too large to host online, is available on request from the lead author.

*Competing interests.* The authors declare no competing interests



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
