# Peer review of "Quantifying and attributing the role of anthropogenic climate change in industrial-era retreat of Pine Island Glacier"

_EGUsphere, 2025_

## Referee Comment (RC1)

**Review of "Quantifying and attributing the role of anthropogenic climate change in industrial-era retreat of Pine Island Glacier" by Alexander Bradley et al.**

This study investigates the long-term (multi-century) historical behavior of Pine Island Glacier (PIG) using a model framework that assimilates sparse but important constraints over this period. The approach enables the authors to calibrate several free parameters, among them aspects of climate forcing, so as to reproduce the known century-scale retreat of PIG. As a result, they can quantify the likely role of anthropogenic trends and a natural early-20th century climate anomaly in driving this retreat. They find from their optimized ("posterior") ensemble of simulations that an anthropogenic trend is necessary to produce the full magnitude of observed retreat, but importantly, major retreat also occurs without an anthropogenic trend.

This study contributes substantially to the line of research attempting to disentangle the drivers of ice loss in the Amundsen Sea region. Notably, while previous studies have advanced understanding of the atmospheric and ocean drivers, the present study is the first to model the ice dynamics of PIG in an attribution framework and thus stands to make an important advance. The simulation + emulation framework seems useful for approaching the considerable uncertainties surrounding this problem, and promising broadly for future work on historical simulations. Overall the study is well-written with clear figures and descriptions of the calibration/emulation methods. And the authors provide transparent discussion of several caveats, which is important and appreciated.

That said, I do have some questions about the experimental setup and results that I think need resolving, and possibly reframing, before publication. I detail these major comments below, as well as some minor and technical comments.

**Major comments**

1) Most generally, I think the attribution assessment should be framed more explicitly as being conditional on the assumption that the glacier is losing mass from 1750 on, regardless of 20th-century forcing. Whether or not this was an explicit assumption at the outset, it is built-in by the combination of the initial condition (which the authors suggest may be too large) and the calibration procedure that is constrained to match the 2015 volume. Consider an alternative possibility that PIG was closer to steady state (i.e., not losing mass) prior to the 1940s event – such a case is plausible given uncertainties in preindustrial conditions, but is not really possible to consider in this framework, by virtue of the initial condition used. And I think that as long as fully communicated, that is fine – it is still valuable to explore the family of possibilities stemming from this initial condition, and the authors include thoughtful discussion around the implications of the initial condition (including considering a range of IC's in future work). But, I think this built-in feature of mass loss should be worked more directly into the summary statements around the anthropogenic component of retreat, as it might exclude some plausible scenarios that could yield different attribution numbers. Again, the discussion around caveats is already helpful, but it could be clearer that the particular numbers reported are conditional on this context of significant mass loss for all simulations (Fig 7b really brings this home for me).

2) Related to the initial condition, it seems several aspects of the model setup would cause initialization artifacts that propagate into the simulations, which seems potentially problematic for assessing different drivers of retreat, and possibly for the calibration procedure.

 - As I understand, forward simulations with perturbed parameters are branched from a common 1750 initial condition. I don't see anything about repeating the spinup procedure with the perturbed parameters (If that is what's done, please clarify!). So in the case of temporally-static parameters (viscosity, sliding, and melt-rate prefactors), the model is responding transiently to a parameter perturbation at 1750 which, given long ice-sheet response times, I'd expect blurs into the effects of climate forcings of interest in the 20th century.
 - Also, the preindustrial state is achieved by shutting off melt for 500 followed by a 50-year spinup with nonzero melt (but cold conditions). 50 years is very little time for the ice dynamics to adjust. Would the GL be stable here indefinitely with the low but nonzero melt rates? (i.e., even before calibrating parameters to match observations). Also, if I'm not mistaken, there is a jump in the pycnocline depth imposed at the start of the historical simulations, from -600 m in the spinup (line 215) to -500 m as the central value in forward simulations (line 208). Does this add another initialization shock?

Overall, these add ambiguity to the trajectories going into the 20th century. How long do these adjustments persist? The authors do suggest that longer spinups would be desirable, but also that 1750 is a sufficient start time (L 559). It would be helpful to discuss the rationale in more detail, and more ideally to provide some sort of control simulation to help characterize such transient effects.

3) I also wonder about the plausibility of the climate forcing inferred by these simulations. I see the value in the approach to treat $B_0$ and $T_0$ as free parameters in the calibration framework (and then removing them in counterfactual scenarios). However, the posterior values for both $B_0$ and $T_0$ end up as very strong forcings. Again, to the authors' credit, this is clearly pointed out, but I think some more discussion is warranted.

In particular, the posterior $B_0$ of > 200 m is quite large compared to the imposed stochastic anomalies. I'll note the example in Fig. 2 ($B_0$ = 50 m) is not very representative: $B_0$ = 200 m would be literally off the chart. Is this really realistic? The posterior $T_0$ of ~ 200 m/century is also quite large compared to the example shown in Fig 2. The authors provide helpful references for comparison, but still, both $B_0$ and $T_0$ seem larger in a signal-to-noise sense than the corresponding event / trends in reconstructions that are cited (e.g., O'Connor et al 2023; Naughten et al. 2024). Granted these references are providing different variables (winds, sea level pressure, subsurface temp), but it's hard to imagine the signal-to-noise would be greatly amplified for pycnocline depths. Alternatively, perhaps the stochastic variability applied here is too small.

Either way, I think it would be important for the authors to clarify further how plausible they expect these posterior values are. Given that the model calibration process seems to push

all parameters towards values that enable retreat, it seems it may be selecting for forcings that are stronger than realistic. (On the other hand, it is interesting that despite these strong inferred forcings, there is still large-scale retreat in the counterfactual simulations without them.) It is concluded that anthropogenic forcing contributes ~20% of the observed retreat, but if either $B\_0$ and/or $T\_0$ are on the high end of what's plausible, that would seem to be an important caveat. I also think it is important to show the reader an example of an actual posterior pycnocline timeseries (as opposed to the example in Fig. 2c) so it is clear what magnitudes of forcing are implied here.

**Minor and technical comments**

- Abstract (and elsewhere) – I'm not sure how directly the Holocene retreats should be invoked here, since they aren't directly addressed in the present study. I think it is a great discussion point, but even though the simulations here start off with mass loss in 1750 so are consistent with this idea, the residual of multi-millennium retreat is not being directly simulated here, so I would consider qualifying how this possibility is raised.
- 135 – "multiplies the…" (something is missing)
- 150 – How does this set value for C compare to the areas where it is inferred?
- Fig 1c – is the preindustrial profile shown that without melt, or after the melt is re-introduced? (And especially if the latter, I'm not sure it should be referred to as a steady state.
- 210 – It would be helpful to specify more about the stochastic variability imposed. Specifically, is it Gaussian-distributed and just truncated at -2, 2? If so, what does alpha correspond to in terms the distribution? (I'm guessing 4*sigma, but it should be specified). Also note the timescale of the autoregressive process.
- 222 – given that the 1940s event is associated with internal climate variability, it seems somewhat inconsistent to superimpose a representation of it on top of stationary stochastic anomalies. So, those realizations with positive R(t) anomalies around 1940 will add further to $B\_0$, creating a double(ish) anomaly? I understand the rationale for directly imposing the 1940s event, but perhaps the reader should be alerted to this.
- 270 – is there a source for this error estimate, or just an order-of-magnitude estimate?
- Table 1 – specify units for $T\_0$ – meters per century?
- Fig 3 – specify – results are shown for a single realization of stochastic forcing? (To disambiguate from different iterations of the imposed event/trend forcings).
- ~285-295 – check case for thetas – it seems inconsistent. Or is capital vs. small theta supposed to mean something?
- 343 – interesting that using all iterations improves emulator performance – is this because it's still a valid mapping between parameters and model state? (and matching observations isn't important for training emulators?)
- 345 – I'm confused by the notion of coverage here – isn't the percentage of emulator predictions falling outside 2 stdv just defined by stdv? Or is this comparing two

distributions? Is the emulator standard deviation defined by the analytic uncertainty estimates mentioned earlier?

- 370 and on – I appreciate the plain-language descriptions alongside the more formal descriptions – I think this will be helpful for readers.
- 405 – perhaps "particular" rather than "precise"?  The latter seems inconsistent when followed with "broad errors on observational constraints"
-  438 – this follows my major comment above, but I'm skeptical that the CES procedure here should be taken as inferring a lot about the climate forcing. Around line 423, it is noted that via the melt prefactor, the procedure is causing higher melt rates than observed. It seems like the constraint to make the model lose a lot of mass by 2015 might also be biasing the B_0 to be high.
- 440 – I would clarify "the *full magnitude of* the 20th century retreat…" as is done elsewhere. Here, it could be construed as no retreat occurring without anthropogenic forcing, which is not what is found.
- 515 and on – it is quite an interesting finding that there are points where the fraction of attributable retreat decreases when the grounding lines across scenarios are pinned at the same bedrock highs. It makes me wonder whether a fraction of attributable volume loss would show the same? It seems the anthropogenic warming could drive more mass loss during the periods of retreat, though I'm not sure of my intuition here.
- 523 – when providing these projections, perhaps remind the reader this is subject to the extending the simplified forcing scenario. It seems this could vary across future projections.
- 553 – I'm unclear on what is meant by pre-1940s forcing being too week. Baseline melt rates, or the stochastic variability?
- 593 – That ice-shelf area change is not included in the model seems significant, and should probably be noted much earlier, in the model description. What are the implications of this? Presumably there is still an effect on buttressing through ice-shelf thinning, right?
- 601 – again, I suggest specifying "attribute the component of retreat due to…"  since the full magnitude is not found to be attributable to anthropogenic forcing.

**References**

Naughten, K.A., Holland, P.R. & De Rydt, J. Unavoidable future increase in West Antarctic ice-shelf melting over the twenty-first century. *Nat. Clim. Chang.* **13**, 1222–1228 (2023). https://doi.org/10.1038/s41558-023-01818-x

O'Connor, Gemma K., et al. "Characteristics and rarity of the strong 1940s westerly wind event over the Amundsen Sea, West Antarctica." *The Cryosphere* 17.10 (2023): 4399-4420.

---

## Author Comment (AC1)

**Response to reviewer 1**

We would like to thank the referee for their detailed review of our manuscript. Responding to these comments forced us to think hard about our manuscript and make a number of improvements, particularly around the framing of the work. We believe that these changes have improved both the clarity and consistency of our work. The reviewer's original comments are in black, with our responses in blue. In places where we refer back to earlier comments in the responses, we have labelled comments "[reference comment x]".

This study investigates the long-term (multi-century) historical behavior of Pine Island Glacier (PIG) using a model framework that assimilates sparse but important constraints over this period. The approach enables the authors to calibrate several free parameters, among them aspects of climate forcing, so as to reproduce the known century-scale retreat of PIG. As a result, they can quantify the likely role of anthropogenic trends and a natural early-20th century climate anomaly in driving this retreat. They find from their optimized ("posterior") ensemble of simulations that an anthropogenic trend is necessary to produce the full magnitude of observed retreat, but importantly, major retreat also occurs without an anthropogenic trend.

This study contributes substantially to the line of research attempting to disentangle the drivers of ice loss in the Amundsen Sea region. Notably, while previous studies have advanced understanding of the atmospheric and ocean drivers, the present study is the first to model the ice dynamics of PIG in an attribution framework and thus stands to make an important advance.

The simulation + emulation framework seems useful for approaching the considerable uncertainties surrounding this problem, and promising broadly for future work on historical simulations. Overall the study is well-written with clear figures and descriptions of the calibration/emulation methods. And the authors provide transparent discussion of several caveats, which is important and appreciated.

That said, I do have some questions about the experimental setup and results that I think need resolving, and possibly reframing, before publication. I detail these major comments below, as well as some minor and technical comments.

Major comments

1) Most generally, I think the attribution assessment should be framed more explicitly as being conditional on the assumption that the glacier is losing mass from 1750 on, regardless of 20th-century forcing. Whether or not this was an explicit assumption at the outset, it is built-in by the combination of the initial condition (which the authors suggest may be too large) and the calibration procedure that is constrained to match the 2015 volume. Consider an alternative possibility that PIG was closer to steady state (i.e., not losing mass) prior to the 1940s event – such a case is plausible given uncertainties in preindustrial conditions, but is not really possible to consider in this framework, by virtue of the initial condition used. And I think that as long as fully communicated, that is fine – it is still valuable to explore the family of possibilities stemming from this initial condition, and the authors include thoughtful discussion around the implications of the initial condition (including considering a range of IC's in future work). But, I think this built-in feature of mass loss should be worked more directly into the summary statements around the anthropogenic component of retreat, as it

might exclude some plausible scenarios that could yield different attribution numbers. Again, the discussion around caveats is already helpful, but it could be clearer that the particular numbers reported are conditional on this context of significant mass loss for all simulations (Fig 7b really brings this home for me).

We agree with this very good point raised by the reviewer and have made the attribution assessments more clearly conditional on this choice of initial state (details below). In particular, we have made our assumption that the ice state in the pre-industrial is that following the cold forcing spin-up clearer and we have changed our wording to clarify, referring to it as the 'initial state' rather than 'pre-industrial' state – in this way we disentangle our model assumption ("initial") from the real world ("pre-industrial").

In the updated manuscript, we explicitly state that the attribution assessments are conditional on the choice of initial state, writing in the abstract:

*"These results are, importantly, conditional on our choice of initial state. For our chosen initial state, we find that the parameter combinations compatible with these observational constraints require PIG to lose mass (but not experience grounding line retreat) over the entire simulated period since 1750, not just after the 1940s when grounding line retreat was initiated. This preconditioned ice mass loss introduces significant uncertainty into our quantification of 20th century forcing contributions."*

We have also made the initial state assumption clearer when we introduce the model setup, writing:

*"The ice geometry after the spin-up is considered to be a reference pre-industrial state, and all results presented here are conditional on this choice. In our framework, it is not a true steady state for each simulation because we vary the model parameters at the start of the simulation (see Section 2.1.3) and the ice geometry will respond to this perturbation. The time that this takes depends on the specific model parameter perturbation; it is hard to assess a priori, as the CES procedure itself selects model parameter values which are compatible with observations. For the prior-mean parameter combinations, this response time is small but, as we shall see, the CES procedure nudges parameter distributions some way from the prior means."* [reference comment 1]

And we have made it clearer in the results section, writing:

*In our statistical reconstruction, the grounding line remains pinned on the ridge until the 1940s, when retreat is initiated (figure 7a). This is consistent with the imposed observational constraint on grounding line position in the 1940s. The grounded volume, however, decreases from the beginning of the simulation, with accelerating retreat beginning in the 1960s (figure 7b). The parameters selected by the procedure, which result from our experimental design (in particular, our three observational controls, combined with choice of initial state), effectively precondition the ice sheet to lose volume, but not experience grounding line retreat, from the start of the simulation. Without a pre satellite-era control on ice volume, it is difficult to ascertain whether, in reality, the ice sheet was losing mass over this entire period or not. An alternative scenario is that the ice volume was close to steady state for hundreds to thousands of years prior to the satellite record, but that scenario cannot be considered by this analysis. Given that the grounding line position is stable, but the ice volume decreases prior to the 1940s, the ice sheet*

*is adjusting during this period to the change in parameter values imposed on it at the start of the simulation (the shocks discussed in Section 2). We do not know in advance how long these adjustment periods will be, because we do not know which parameters will be selected by the CES procedure. However, for those model parameters selected here, it appears to continue for some time into the simulation. Therefore, the effect of trends in forcing are likely obscured by these initial condition effects. In what follows, we therefore report all attribution results as conditional on our choice of initial state. We discuss possible solutions to this problem in the discussion."* [reference comment 2]

We have also expounded upon our discussion around solutions to this initial state problem in the updated manuscript, writing:

*"To address these potentially important initialization issues, we recommend that the initial state and historical forcing be treated as additional parameters to be constrained by the CES procedure. The former could be achieved, for example, by scaling the size of the ice volume by a prefactor (we effectively took a prior distribution on this parameter to be a delta function, with no uncertainty), or perhaps by including an extra parameter that defines a shift to the initialization date. More generally, our results highlight the importance of selecting an appropriate initial state when simulating retreat of the Antarctic Ice Sheet."*

We are keen to stress that (and the referee is aware of this) we have not explicitly imposed that PIG is losing mass in 1750, but rather have implicitly done so, by imposing the initial state and then, to match the observational constraints, the simulations lose mass immediately from 1750. However we do not know this will happen in advance – it emerges from the procedure: it could have been instead that the mass loss was not initiated until the 20[th] century climatic forcing has taken effect. This distinction is important because (1) we are using a Bayesian framework, in which our assumptions should be made explicit at the outset and (2) in terms of the grounding line position, the ice does not retreat (i.e. retreat is not necessarily baked in by the assumption of the initial state). We make a note of this distinction in the updated manuscript.

2) Related to the initial condition, it seems several aspects of the model setup would cause initialization artifacts that propagate into the simulations, which seems potentially problematic for assessing different drivers of retreat, and possibly for the calibration procedure.

- As I understand, forward simulations with perturbed parameters are branched from a common 1750 initial condition. I don't see anything about repeating the spinup procedure with the perturbed parameters (If that is what's done, please clarify!). So in the case of temporally-static parameters (viscosity, sliding, and melt-rate prefactors), the model is responding transiently to a parameter perturbation at 1750 which, given long ice-sheet response times, I'd expect blurs into the effects of climate forcings of interest in the 20th century.

This is a very good point, which we failed to touch upon in our original manuscript. We use the same initial condition for each parameter combination and make this more explicit in the updated manuscript, writing:

*"Each of our simulations begins in 1750, using the same initial state."*

The response time of the model to the changing of the parameters at the start of the simulation will depend strongly on the specific parameter variations. For some choices of parameters, the ice volume will reach steady state over the 200 years prior to anthropogenic forcing kicking in, while for others it will take much longer. In an ideal (computationally infinite) world, we would spin up the model over many centuries (or millennia) for each parameter combination, but this is restricted by computational expense. We compromise on this by running the simulations from 1750, but the memory of this may be long, as the reviewer points out. We make this clearer in the updated manuscript – see reference comments 1 and 2 above.

We also add a note of this adjustment time, writing:

*"Each of our simulations begins in 1750, using the same initial state. Ideally, we would run our simulations for longer, by setting the initial state to correspond to further back in time, allowing memory of the initial state to be lost and removing the effect of the `initialisation shock' described above. However, this incurs an additional computational expense, and 1750 is chosen as a balance between extra computational expense and attempting to run for long enough to minimise initial state dependence."*

As noted above, we have also made the implications of these initialization shocks clear in the results section – see reference comment 2.

- Also, the preindustrial state is achieved by shutting off melt for 500 followed by a 50 year spinup with nonzero melt (but cold conditions). 50 years is very little time for the ice dynamics to adjust. Would the GL be stable here indefinitely with the low but nonzero melt rates? (i.e., even before calibrating parameters to match observations).

Due to only low melt rates from the applied cold conditions, and no grounding-line displacement, the ice dynamics have largely adjusted after the 50 year spin-up and neither the ice volume, nor grounding line position, vary significantly for a longer spin-up. However, as we outline above, the posterior-mean parameter values are some way from the prior-mean values for certain parameters, and this is why the initialization shocks may be important in the results.

Also, if I'm not mistaken, there is a jump in the pycnocline depth imposed at the start of the historical simulations, from -600 m in the spinup (line 215) to -500 m as the central value in forward simulations (line 208). Does this add another initialization shock?

This is another good point, about which we did not include sufficient detail in the original manuscript. We agree that, as for the change in model parameters, this will introduce an initialization shock; the time persistence of this shock will, again, depend on the choice of model parameters and is therefore difficult to quantify, but may also obscure the role of 20[th] century trends in forcing if it extends beyond 1940s. In the updated manuscript, we have made this additional shock more explicit in the model description, writing:

*"In the spin-up period described in the previous section, the pycnocline centre is set to a constant value of -600m. This is to ensure that the ice sheet is in a pseudo-steady state after the spin-up period, for those parameters used for the spin up. When the stochastic forcing (equation (5)) is turned on, the mean pycnocline centre becomes -500m. The ice sheet will respond to this perturbation at the start of the simulation (in addition to responding to the*

*changes in ice sheet model parameters, including those that relate to ice shelf melting, that take place at the same time)."*

In the discussion section of the updated manuscript, we make it clear that these initialization shocks (from model parameters and change in forcing), may blur into the trends, writing:

*"This picture is confused by initialization shocks - necessary step changes in the forcing and model parameters at the start of the simulations; the model's response to these changes appears to blur into the effects of climate forcings of interest in the 20th century."*

Overall, these add ambiguity to the trajectories going into the 20th century. How long do these adjustments persist? The authors do suggest that longer spinups would be desirable, but also that 1750 is a sufficient start time (L 559). It would be helpful to discuss the rationale in more detail, and more ideally to provide some sort of control simulation to help characterize such transient effects.

Regarding longer spin-ups: we apologise for our imprecise wording – what we meant is that we include the spin-up as an attempt to counter some of the initialization issues outlined above. In the updated manuscript, we have clarified this, writing:

*"1750 is chosen as a balance between extra computational expense and attempting to run for long enough to minimise initial state dependence."*

As we outline above, putting a precise timescale on the adjustments is not well-defined because it depends on the choice of model parameters. For the default parameters, this adjustment timescale is short, but for other parameter combinations it may be much longer (including extending beyond the 1940s) and we don't know in advance which parameters will be selected by the procedure.

As discussed, it is difficult to quantify the role of these effects and is not clear what a control simulation would involve. However, we hope that the changes outlined above make it clear that these effects are playing some role. What we can say is that the overall response is a combination of a contribution from the initialization shock and a contribution from $20^{th}$ century forcing components (the 1940s event and the anthropogenic trends), but we cannot disentangle these with our setup. If we removed the effects of initialisation shock (for example, by allowing the procedure to also pick the initial state, as outlined above), then the response would be purely a result of the $20^{th}$ century forcing.

These effects introduce significant uncertainty into our quantification of the role of anthropogenic trends in forcing. For the attribution assessments, this is a crucial point and we make this clear in the updated manuscript: writing, firstly, in the abstract:

*"This preconditioned ice mass loss introduces significant uncertainty into our quantification of 20th century forcing contributions."*

And, secondly, in the discussion:

*"These results are conditional on our choice of initial state, which, after having completed the CES procedure, imposes that PIG must lose mass consistently since 1750 to reproduce the observed ice volume in 2015. The overall response is a combination of a contribution from the initialization shock and a contribution from 20ᵗʰ century forcing components (the 1940s event and the anthropogenic trends), but we cannot disentangle these with our setup. If we removed the effects of initialisation shock (for example, by allowing the procedure to also pick the initial state, as outlined above), then the response would be purely a result of the 20ᵗʰ century forcing or long-term thinning trends. We therefore conclude that the precise quantified role of 20ᵗʰ century forcing components is highly uncertain."*

I also wonder about the plausibility of the climate forcing inferred by these simulations. I see the value in the approach to treat $B_0$ and $T_0$ as free parameters in the calibration framework (and then removing them in counterfactual scenarios). However, the posterior values for both $B_0$ and $T_0$ end up as very strong forcings. Again, to the authors' credit, this is clearly pointed out, but I think some more discussion is warranted.

In particular, the posterior $B_0$ of > 200 m is quite large compared to the imposed stochastic anomalies. I'll note the example in Fig. 2 ($B_0 = 50$ m) is not very representative: $B_0 = 200$ m would be literally off the chart. Is this really realistic? The posterior $T_0$ of ~ 200 m/century is also quite large compared to the example shown in Fig 2. The authors provide helpful references for comparison, but still, both $B_0$ and $T_0$ seem larger in a signal-to-noise sense than the corresponding event / trends in reconstructions that are cited (e.g., O'Connor et al 2023; Naughten et al. 2024). Granted these references are providing different variables (winds, sea level pressure, subsurface temp), but it's hard to imagine the signal-to-noise would be greatly amplified for pycnocline depths. Alternatively, perhaps the stochastic variability applied here is too small.

Either way, I think it would be important for the authors to clarify further how plausible they expect these posterior values are. Given that the model calibration process seems to push all parameters towards values that enable retreat, it seems it may be selecting for forcings that are stronger than realistic. (On the other hand, it is interesting that despite these strong inferred forcings, there is still large-scale retreat in the counterfactual simulations without them.) It is concluded that anthropogenic forcing contributes ~20% of the observed retreat, but if either $B_0$ and/or $T_0$ are on the high end of what's plausible, that would seem to be an important caveat. I also think it is important to show the reader an example of an actual posterior pycnocline timeseries (as opposed to the example in Fig. 2c) so it is clear what magnitudes of forcing are implied here.

The referee makes a good point, which we had not addressed in sufficient detail in the original manuscript.

Firstly, in the updated manuscript, we have added a plot showing the contributions to forcing from the posterior trend and 1940s event (figures 5g and h, respectively).

Secondly, regarding the magnitude of stochastic variability: as outlined in the paper, stochastic variability in pycnocline depth is consistent with the range that is observed (Dutrieux et al., 2014; Webber et al., 2013). We have assumed that the magnitude of stochastic variability in the past is equal to that in the present day observations (and these are late summer only), but,

without oceanographic observations prior to the 1990s and outside the late summer window, we cannot constrain this. We note this in the updated manuscript, writing:

*"The second term on the right-hand side of (1) corresponds to internal variability other than the 1940s event (see below). The shallowest and deepest observed pycnocline depths are approximately -400m and -600m, respectively (Dutrieux et al., 2014) and thus the parameter \alpha = 200m is the magnitude of differences between warmest and coldest conditions associated with internal variability."*

Thirdly, regarding the posterior trend in forcing: we believe that the posterior estimates we find are within the ranges of signal-to-noise ratios of the available reconstructions of other climate variables. In more detail, our posterior mean trend in forcing per century is approximately twice the amplitude of internal variability, which is in the range of other reconstructions (albeit with the caveat that, as the reviewer points out, linear trends in winds etc do not directly correspond to linear trends in pycnocline depth, and these reconstructions include centennial scale internal variability which is not captured by our framework): in Holland et al. (2019), centennial trends in the winds are approximately equal to the amplitude of internal variability (their figure 3a). In Holland et al. (2022), trends in the shelf winds are approximately twice the amplitude of internal variability (their figure 4c). In Naughten et al. (2022) and Naughten et al., (2023), historical anthropogenic trends in ocean conditions are approximately equal to the amplitude of internal variability, while future trends from Naughten et al., (2022) are approximately four times the amplitude of internal variability, regardless of scenario (their figure 3). We have added a note of this in the results section, writing:

*"The posterior centennial trend in forcing is approximately double the magnitude of internal variability (100m). This signal-to-noise ratio is within the range of reconstructions and projections relevant to the region: Holland et al., (2019) report centennial trends in westerly winds approximately equal to the amplitude of internal variability; Holland et al., (2023) report trends in shelf winds which are approximately twice the amplitude of internal variability; and Naughten et al., (2022) and Naughten et al., (2023) report centennial historical trends in ocean temperature which are approximately equal to the amplitude of internal variability, while future centennial trends from Naughten et al., (2023) are approximately four times the amplitude of internal variability, regardless of emissions scenario."*

The referee is correct to point out that the posterior 1940s event corresponds to very strong forcing for that period of time, and indeed that can be seen in the new figure 5h. We are keen to stress that this is what the machinery of the procedure picks out: given our experimental setup and model, it is very likely that there was a large 1940s event (this is also true for the trend in forcing). This provides evidence, from an ice dynamics perspective, that the 1940s event may have been larger than we previously understood. However, this result is likely influenced by either (1) our experimental setup including our initial state or (2) our model (or both), on which this result is conditional. For (1), an important factor may be our choice of prior for the 1940s event; this turns out to be quite similar to the posterior, potentially preconditioning for a large 1940s event. For (2), it may be that our model is not as sensitive to climate forcing as ice sheets are in practice, and therefore extremely large 1940s forcing is required to initiate retreat. There are many reasons why this could be (which we noted in the original manuscript), including fixed ice fronts, unresolved processes in our model, and constant accumulation. We note these points in the discussion of the updated manuscript, writing:

*"The posterior 1940s event is outside the range of imposed internal variability in the model and warmer than any event in the observational record (Dutrieux et al., 2014). Although the 1940s event was exceptional in the context of centennial events, as it stands out in regional climate proxies (Schneider et al., 2008, O'Connor et al., 2023) and induced substantial ice sheet change (Smith et al., 2017, Clark et al., 2024), the values inferred here are large even in this context. There are several possible explanations for this: firstly, it may well be that the 1940s event was indeed larger than previously understood (noting that previous work has focussed on non-oceanographic climate variables), which the present study evidences from an ice dynamics perspective. Secondly, it may suggest that our model is not sufficiently sensitive to climate forcing (for the reasons mentioned above) and therefore a very large anomaly is required to initiate retreat. Finally, our choice of prior 1940s event, which includes a large anomaly, may have shifted posterior values towards a large 1940s event."*

Minor and technical comments

- Abstract (and elsewhere) – I'm not sure how directly the Holocene retreats should be invoked here, since they aren't directly addressed in the present study. I think it is a great discussion point, but even though the simulations here start off with mass loss in 1750 so are consistent with this idea, the residual of multi-millennium retreat is not being directly simulated here, so I would consider qualifying how this possibility is raised.

The referee is right that the connection to the Holocene is perhaps too remote to warrant inclusion in the abstract. What we really mean is (as outlined above) that the initial condition is important and it could be that (1) we have overestimated volume in the initial state/there is initialisation shock or (2) we got the initial state right and so ice sheet retreat over the Holocene, which put the ice in that position, preconditioned the retreat. In the updated manuscript, we remove reference to the Holocene in the abstract, writing instead:

*"…or may suggest that the earlier ice state preconditioned the industrial era retreat, possibly implicating longer term changes to WAIS in the present retreat."*

- 135 – "multiplies the…" (something is missing)

Thanks for spotting, we have removed this superfluous clause in the updated manuscript.

- 150 – How does this set value for C compare to the areas where it is inferred?

As we show in the figure below, this value of C is comparable to the inferred areas (the black solid line is the value of 10,000 Pa m^(-1/3) a^(1/3) that we take). We have also noted that future studies might wish to treat the grounded prefactor and "ungrounded" basal sliding field as independent parameters to be tuned:

*"Note that the basal sliding prefactor also pre-multiplies the drag in these areas and therefore the basal shear stress there is also tunable, however these must be co-varied; since basal drag exerts an important control on ice dynamics, future studies may wish to treat the grounded and ungrounded basal drag coefficient as independent tunable parameters. This value is an order of*

*magnitude estimate (it is the 23rd percentile of the values in the inferred areas from the inversion)."*

[Figure]

- Fig 1c – is the preindustrial profile shown that without melt, or after the melt is re-introduced? (And especially if the latter, I'm not sure it should be referred to as a steady state.)

The reviewer is correct to point this out. Shown in figure 1c is the preindustrial profile after the melt spin up. We have amended the caption to reflect this, writing

*"in the initial state (after the cold forcing spin-up)"*

As we have outlined above, in the updated manuscript, we also state more explicitly (in the model description section) that we are effectively imposing that the ice volume in 1750 to be that from the cold forcing spin-up, and that is what we mean by the 'pre-industrial' state, writing:

*"The ice geometry after the spin-up is considered to be the pre-industrial state, and referred to as the initial state….Each of our simulations begins in 1750, using the same initial state."*

- 210 – It would be helpful to specify more about the stochastic variability imposed. Specifically, is it Gaussian-distributed and just truncated at -2, 2? If so, what does alpha correspond to in terms the distribution? (I'm guessing 4*sigma, but it should be specified). Also note the timescale of the autoregressive process.

Thanks for this suggestion. In the updated manuscript, we have added details on the distribution of the noise, truncation, and timescale of the autoregressive process. We write:

*"R(t) is a dimensionless timeseries generated from a modified first-order autoregressive process: it is as in Christian et al., (2022) and Bradley et al., (2024), with interdecadal-to-decadal timescales well represented, but is truncated between -2 and 2 (so \alpha is four times the standard deviation of this autoregressive process). Noise in the autoregressive process has*

*a Gaussian distribution. We take a decorrelation timescale of 10 years in the autoregressive process, which ensures that it captures decadal variability."*

- 222 – given that the 1940s event is associated with internal climate variability, it seems somewhat inconsistent to superimpose a representation of it on top of stationary stochastic anomalies. So, those realizations with positive R(t) anomalies around 1940 will add further to B_0, creating a double(ish) anomaly? I understand the rationale for directly imposing the 1940s event, but perhaps the reader should be alerted to this.

We agree that imposing the 1940s event may feel a little bit contrived, given that it is technically part of the internal variability. However, as the referee points out, given its important role in the history of the PIG retreat and, that we wish to quantify its effect, it is necessary to include externally to the stochastic anomalies. In addition, we average over realisations of the AR noise, which therefore cancel out (this can be seen in figure 2b). The machinery we use cannot pick a specific climate realization, so there is effectively there is no variability on average and the 1940s event can't exist unless we explicitly impose it.

In the updated manuscript, we explicitly mention that the 1940s event is technically part of the internal variability, to alert the reader, writing:

*"It is important to stress that the 1940s event is technically part of the internal variability, viewed as independent of anthropogenic influence. We choose to impose it in this deterministic (rather than stochastic) way to enable us to quantify its effect on 20th century PIG retreat."*

- 270 – is there a source for this error estimate, or just an order-of-magnitude estimate?

This is an order of magnitude estimate, which we have clarified in the updated manuscript, writing:

*"The observational error on the 2015 grounded volume is taken to be $10^{12}$ m^3. This is a conservative order of magnitude estimate (0.2% of the ice volume in the initial state), based on errors in mass balance estimates from IMBIE (2023), which are on the order of millimetres of sea level rise, while PIG contains order meters of sea level rise potential."*

Note that in the original manuscript we incorrectly quoted a value of 1% of the ice volume (which would be $4.9 * 10^{12}$ m^3), based on an earlier version of the analysis. We have corrected this in the updated manuscript.

- Table 1 – specify units for T_0 – meters per century?

Thanks for spotting this error. We have corrected it in the updated manuscript.

- Fig 3 – specify – results are shown for a single realization of stochastic forcing? (To disambiguate from different iterations of the imposed event/trend forcings).

In the updated manuscript, we have specified in the caption to figure 3 that the results are for a single realization of stochastic forcing:

*"Results are shown for a single realization of forcing…(c)--(h) Scatter plots of grounded ice volume (grv) in 2015 as a function of (c)--(e) climate and (f--h) model parameters for simulations whose trajectories are shown in (a)--(b) (and are therefore only for a single realization of forcing)"*

- ~285-295 – check case for thetas – it seems inconsistent. Or is capital vs. small theta supposed to mean something?

Thanks for spotting this – these should have been capitalised and this has been corrected in the updated manuscript.

- 343 – interesting that using all iterations improves emulator performance – is this because it's still a valid mapping between parameters and model state? (and matching observations isn't important for training emulators?)

Training of the emulator is invariant to the observational constraints, which enter only via the calibration and sampling steps: the emulator simply tries to reproduce the map between input parameters and simulated model state. To that end, the emulator doesn't know anything about the observations and emulator performance: its performance is only assessed by how well it can reproduce the *simulated* values of volume and grounding line position, rather than how well it reproduces the *observed* values. Including all iterations, rather than only the final one, means that there is more training data, which is why the emulator performance is improved in this case. We appreciate this is a potentially confusing area, and so we have clarified this in the updated manuscript, writing:

*"We train the emulators on all of the simulations from the EKI, rather than following the suggestion of Cleary et al., 2021 to just use those from the final iteration, as this was found to improve emulator performance, i.e. the ability of the emulators to reproduce the simulated model output. This is likely because using all iterations means that the training data set (the set of simulated model outputs and corresponding input parameters \Theta) is larger."*

- 345 – I'm confused by the notion of coverage here – isn't the percentage of emulator predictions falling outside 2 stdv just defined by stdv? Or is this comparing two distributions? Is the emulator standard deviation defined by the analytic uncertainty estimates mentioned earlier?

For a given set of input parameter values (\Theta), the emulator (a Gaussian process) returns a Gaussian distribution of predictions of the model output, which is characterised by a mean (the central emulator prediction) and a standard deviation. It is this standard deviation of these Gaussian distributions that we refer to when we talk about the emulator errors. This is rather than the standard deviation of the set of all central emulator predictions, which we believe the reviewer is referring to. We appreciate that this is a subtle point and have therefore clarified in the updated manuscript, writing:

*"For a given set of inputs, a Gaussian process returns a Gaussian distribution of outputs, which is characterised by its mean and standard deviation. The mean and standard deviation can be considered the emulator's central estimate of the model output and its uncertainty in this prediction, respectively, for these parameter values."*

We have also clarified this further when discussing the coverage, writing:

*"…the best coverage (the percentage of emulator predictions that lie more than two emulator standard deviations away from corresponding model output)"*

- 370 and on – I appreciate the plain-language descriptions alongside the more formal descriptions – I think this will be helpful for readers.

Thank-you!

- 405 – perhaps "particular" rather than "precise"? The latter seems inconsistent when followed with "broad errors on observational constraints"

Thanks for this suggestion – we have adopted this wording in the updated manuscript.

- 438 – this follows my major comment above, but I'm skeptical that the CES procedure here should be taken as inferring a lot about the climate forcing. Around line 423, it is noted that via the melt prefactor, the procedure is causing higher melt rates than observed. It seems like the constraint to make the model lose a lot of mass by 2015 might also be biasing the $B_0$ to be high.

We have responded to this comment in the "major comments" above.

- 440 – I would clarify "the full magnitude of the 20th century retreat…" as is done elsewhere. Here, it could be construed as no retreat occurring without anthropogenic forcing, which is not what is found.

This is a good point. We have adopted this wording in the updated manuscript.

- 515 and on – it is quite an interesting finding that there are points where the fraction of attributable retreat decreases when the grounding lines across scenarios are pinned at the same bedrock highs. It makes me wonder whether a fraction of attributable volume loss would show the same? It seems the anthropogenic warming could drive more mass loss during the periods of retreat, though I'm not sure of my intuition here.

This is indeed an interesting and subtle point. This can be seen in the trajectories themselves (figure 8a): the fraction of attributable retreat decreases when grounding lines are pinned basically because the trajectories with no trend have time to "catch up" in the retreat at the points. In the case with an anthropogenic trend, the simulations not only reach the pinning points first, but also persist at them for shorter durations. This is similar to Bett et al. (2024), who showed that a significant portion of retreat comes down to how long ice shelves persist on local bedrock highs, with warm scenarios simply removing contact with these highs more quickly.

In our framework, the fraction of attributable volume loss is not well-defined because we don't have a "pre-industrial control" (i.e. an observation of ice volume prior to the anthropogenic trends in forcing beginning. However, we expect that it would be as the reviewer suggests: that the anthropogenic forcing drives more loss during periods of retreat.

We have added a note of this subtlety in the updated manuscript, writing:

*"Essentially the fraction of attributable retreat is controlled by the persistence time at pinning-points, with anthropogenic trends in forcing leading to removal from these points sooner, as has been demonstrated to be a leading order control on future WAIS retreat rates (Bett et al., 2024)."*

- 523 – when providing these projections, perhaps remind the reader this is subject to the extending the simplified forcing scenario. It seems this could vary across future projections.

This is a good reminder for us – thank you. In the updated manuscript, we have added a further note of this assumption to the end of the paragraph in question, writing:

*"We stress, however, that these projections are conditional on the idealised forcing scenario that the linear trend in anthropogenic forcing continues indefinitely."*

- 553 – I'm unclear on what is meant by pre-1940s forcing being too weak. Baseline melt rates, or the stochastic variability?

We have removed this comment in the updated manuscript to avoid confusion, but to clarify: the (non 1940s-event) internal variability component of the forcing has the pycnocline oscillating between present day observed values at all times, not just in the present day. In practice, because our simulations find there has been an anthropogenic trend in forcing, these values imposed prior to 1930 should have been lower (though we do not know this in advance of performing the simulations).

- 593 – That ice-shelf area change is not included in the model seems significant, and should probably be noted much earlier, in the model description. What are the implications of this? Presumably there is still an effect on buttressing through ice-shelf thinning, right?

We are keen to stress that the ice shelf area is not fixed but rather the ice front position is – i.e the grounding line can retreat. The ice front position has not changed significantly on a centennial scale. Buttressing can indeed still evolve through ice shelf thinning. In the updated manuscript, we have noted this assumption in the model description, writing:

*"…on floating ice shelves, which have a fixed ice front, indicated in figure 1b. This ice front position is equal to the 2025 ice front position."*

- 601 – again, I suggest specifying "attribute the component of retreat due to…" since the full magnitude is not found to be attributable to anthropogenic forcing.

We have adopted this wording in the updated manuscript.

References
Naughten, K.A., Holland, P.R. & De Rydt, J. Unavoidable future increase in West Antarctic iceshelf melting over the twenty-first century. Nat. Clim. Chang. 13, 1222–1228 (2023). https://doi.org/10.1038/s41558-023-01818-x

O'Connor, Gemma K., et al. "Characteristics and rarity of the strong 1940s westerly wind event over the Amundsen Sea, West Antarctica." The Cryosphere 17.10 (2023): 4399-4420.

---

## Author Comment (AC2)

**Response to reviewer 2**

Review of "Quantifying and attributing the role of anthropogenic climate change in industrial-era retreat of Pine Island Glacier" by Bradley et al

This study applies an uncertainty quantification framework called "calibrate-emulate-sample" to the multi-centennial evolution of Pine Island Glacier. The motivation for the "calibrate-emulate-sample" method is its iterative nature, which results in more members that agree with observations, compared to the standard Latin Hypercube Sampling plus Importance Sampling. The experimental design is very thoughtful, the manuscript is very well written and the authors do an excellent job highlighting potential caveats when necessary. The methods are clearly explained, and I was able to understand the gist of the "calibrate-emulate-sample" framework (I am not an expert in this field and thus not qualified to assess the details of the method).

We are grateful to the referee for their review of our manuscript and pleased to receive a positive review. Here, we respond to their comments in detail.

Main comments:
Focusing only on Figure 7a, one is tempted to conclude that all four ensembles explain the two observations of grounding line position equally well. This makes me wonder how robust your attribution to anthropogenic climate change really is.

While we agree that each of the ensembles reproduces the retreat to some extent, we disagree that they all explain the observations equally well. Indeed, the 18% that we quote through manuscript is the difference between the "all forcings" and "no anthropogenic trend" ensembles, evidencing that there is a difference between them.

Our choice of plotting style in the original manuscript may have obscured the differences between the different ensembles. To enable the differences between the ensembles to be seen more clearly, we have added figures showing mean and standard deviation of grounding line retreat between 1940 and 2015, and grounded volume in 2015 relative to observed (figures 7c and d in the updated manuscript). In these, the differences between the ensembles can be more clearly seen – anthropogenic forcing shifts the distributions to the right (corresponding to larger retreat and higher ice volume loss). We elaborate on this point in the updated manuscript, writing:

*"In figure 7c, we show distributions of grounding line change between 1930 and 2015 in the four ensembles. Only in the all-forcings and no-trend ensembles is the ensemble mean plus one standard deviation within the observational range, accounting for the observational error (i.e. the right hand of the bar for the ensembles lies within the red shaded area). From this we see clearly how both the anthropogenic trend in forcing and 1940s event shift the distribution of grounding line retreat towards enhanced retreat. However, consistent with the retreat of all ensembles, as outlined above, these shifts are not extreme and the distributions still overlap."*

Are there any additional observations that could be used to further constrain the ensembles? (e.g. Sentinel images that provide the front position, observed velocities). How well does your ice sheet model reproduce reality besides retreat, e.g., velocities, dhdt? In addition, mapping

grounding line position onto a center-line is a relatively weak metric. Would you get a different result by using the floating/grounded mask and a Jaquard Score?

There are of course many different (satellite) observations that we could assimilate into the CES procedure. However, most of these are already used in the inversion procedure (including dh/dt), so are already taken into account. In addition, for the CES procedure to work well, all we need are large scale bulk metrics that are well spread over time, to constrain the centennial evolution. Satellite observations are good to constrain the present-day inversion, but weaker for the CES machinery as their changes over the modern observational era are small on the scale of the 1940s-present changes. In addition, including further present day observations into the procedure naturally downweights the 1940s observations which are crucial and only create one data point (we have already prioritised the present day somewhat by including both volume and grounding line position for 2015, versus grounding line only for 1940). We clarify this in the updated manuscript, as well as making a distinction between the spatially varying fields used in the inversion, and temporally varying fields used in the CES machinery. We clarify these points in the updated manuscript:

*"We also made several choices during the CES procedure, which should be noted. Firstly, we chose to use observations of grounding line position in 1930 and 2015, and grounded volume in 2015. There are a wide variety of satellite observational datasets available which we which could have further assimilated into the procedure. However, we elected not to as most of these are already used in the inversion, so are already taken into account. In addition, for the CES procedure to work well in this instance, all we need are large scale bulk metrics that are well spread over time, to constrain the centennial evolution. Satellite observations are good to constrain the present-day inversion, but weaker for the CES machinery as their changes over the modern observational era are small on the scale of the 1940s-present changes. In addition, including further present day observations into the procedure naturally down-weights the 1940s observations which are crucial and only create one data point (we have already prioritised the present day somewhat by including both volume and grounding line position for 2015, versus grounding line only for 1940)."*

- You construct three emulators, one for each target (GL 1930, 2015, Volume 2015). Would your findings change if you used one emulator that predicts all three targets?

The referee raises a good point, which we had not addressed in the original manuscript. We elected to construct different emulators for each of the targets because, when attempting to emulate all three targets simultaneously, we encountered convergence issues during emulator training (specifically, the training process requires us to invert a matrix which is poorly scaled).

It is difficult to assess the sensitivity of our results to the choice of number of emulators because it requires us to re-run the entire set of posterior simulations again. This is because a different choice of emulator(s) will give slightly different posterior distributions of model parameters and thus different samples for the posterior ensembles. Given that our emulators display good performance in regard to RMSE and coverage, we believe it is a good representation of the underlying simulation space and any other choice of emulator(s) with similar performance would yield the same posterior distributions.

The reviewer's comment is a good one and further work should investigate the sensitivity of posterior distributions to the choice of emulator.

We have added a note of these points in the updated manuscript, writing:

*"Secondly, we used Gaussian processes to emulate these observations and, in particular, using individual Gaussian processes for each of the observational constraints. We chose to emulate the observational constraints individually as, when attempting to emulate them all simultaneously, we encountered convergence issues associated arising from the fact that the matrix required to be inverted during the training process was poorly scaled. Our choice of Gaussian processes enables us to obtain uncertainty estimates in emulated values of these quantities, which are propagated through to attribution assessments via posterior distributions of model parameters. Given that our emulator displays good performance, and our results include emulator uncertainties, we do not believe that the results would change if another, different emulator with similar performance was chosen, but future work should investigate the sensitivity of posterior distributions to the choice of emulator."*

Kind regards,
Andy Aschwanden
University of Alaska Fairbanks

Minor comments:

Melt prefactor vs Sliding prefactor. The sampler seems most opinionated about these two parameters, are they strongly anti-correlated?

This is a good point. Below, we include below a scatter plot of the melt prefactor and sliding prefactor from all 1400 simulations in the EKI, with colours corresponding to the dimensionless error in the grounded volume. Visually, there is a weak negative correlation, though the melt prefactor appears a stronger control on the output than the sliding prefactor. This is confirmed by a partial correlation between the two of -0.0668. This correlation is to be expected physically: a higher (lower, respectively) sliding prefactor would reduce (promote) retreat, while a higher (lower) melt prefactor has the opposite effect, promoting (reducing) retreat. We note this in the updated manuscript, writing:

*"The sampler is most opinionated on the sliding prefactor and melt prefactor, and a partial covariance analysis reveals them to be weakly anti-correlated (R = -0.0668); physically this is to be expected: a higher (lower, respectively) sliding prefactor would reduce (promote) retreat, while a higher (lower) melt prefactor has the opposite effect, promoting (reducing) retreat."*

[Figure]

L 135: "which premultiplies the . $A$..." (remove ".")

Thank-you for spotting this typo, we have fixed it in the updated manuscript.

L 219 and 220: There is no Figure 2d, I assume you mean 2c.

Fixed, thanks.

L 343 ...by Cleary et al (2021)...

Fixed, thanks.

L 393 (and elsewhere) "hasn't" -> "has not"

Fixed, thanks.

Figure 3: "...as a function of (c-e) model and (f-h) climate parameters..." I think those are switched, (c-e) are climate and (f-h) are model parameters.

Thanks for spotting this – you are correct. We have fixed this in the updated manuscript.